# Meta-Learning and Universality:
## Deep Representations and Gradient Descent can Approximate any Learning Algorithm

**Chelsea Finn & Sergey Levine**
University of California, Berkeley
{cbfinn,svlevine}@eecs.berkeley.edu

## Abstract

Learning to learn is a powerful paradigm for enabling models to learn from data more effectively and efficiently. A popular approach to meta-learning is to train a recurrent model to read in a training dataset as input and output the parameters of a learned model, or output predictions for new test inputs. Alternatively, a more recent approach to meta-learning aims to acquire deep representations that can be effectively fine-tuned, via standard gradient descent, to new tasks. In this paper, we consider the meta-learning problem from the perspective of universality, formalizing the notion of *learning algorithm approximation* and comparing the expressive power of the aforementioned recurrent models to the more recent approaches that embed gradient descent into the meta-learner. In particular, we seek to answer the following question: does deep representation combined with standard gradient descent have sufficient capacity to approximate any learning algorithm? We find that this is indeed true, and further find, in our experiments, that gradient-based meta-learning consistently leads to learning strategies that generalize more widely compared to those represented by recurrent models.

## 1 Introduction

Deep neural networks that optimize for effective representations have enjoyed tremendous success over human-engineered representations. Meta-learning takes this one step further by optimizing for a learning algorithm that can effectively acquire representations. A common approach to meta-learning is to train a recurrent or memory-augmented model such as a recurrent neural network to take a training dataset as input and then output the parameters of a learner model (Schmidhuber, 1987; Bengio et al., 1992; Li & Malik, 2017a; Andrychowicz et al., 2016). Alternatively, some approaches pass the dataset and test input into the model, which then outputs a corresponding prediction for the test example (Santoro et al., 2016; Duan et al., 2016; Wang et al., 2016; Mishra et al., 2018). Such recurrent models are *universal learning procedure approximators*, in that they have the capacity to approximately represent any mapping from dataset and test datapoint to label. However, depending on the form of the model, it may lack statistical efficiency.

In contrast to the aforementioned approaches, more recent work has proposed methods that include the structure of optimization problems into the meta-learner (Ravi & Larochelle, 2017; Finn et al., 2017a; Husken & Goerick, 2000). In particular, model-agnostic meta-learning (MAML) optimizes *only* for the initial parameters of the learner model, using standard gradient descent as the learner's update rule (Finn et al., 2017a). Then, at meta-test time, the learner is trained via gradient descent. By incorporating prior knowledge about gradient-based learning, MAML improves on the statistical efficiency of black-box meta-learners and has successfully been applied to a range of meta-learning problems (Finn et al., 2017a;b; Li et al., 2017). But, does it do so at a cost? A natural question that arises with purely gradient-based meta-learners such as MAML is whether it is indeed sufficient to only learn an initialization, or whether representational power is in fact lost from not learning the update rule. Intuitively, we might surmise that learning an update rule is more expressive than simply learning an initialization for gradient descent. In this paper, we seek to answer the following question: does simply learning the initial parameters of a deep neural network have the same representational power as arbitrarily expressive meta-learners that directly ingest the training data at meta-test time? Or, more concisely, does representation combined with standard gradient descent have sufficient capacity to constitute any learning algorithm?

We analyze this question from the standpoint of the universal function approximation theorem. We compare the theoretical representational capacity of the two meta-learning approaches: a deep network updated with one gradient step, and a meta-learner that directly ingests a training set and test input and outputs predictions for that test input (e.g. using a recurrent neural network). In studying the universality of MAML, we find that, for a sufficiently deep learner model, MAML has the same theoretical representational power as recurrent meta-learners. We therefore conclude that, when using deep, expressive function approximators, there is no theoretical disadvantage in terms of representational power to using MAML over a black-box meta-learner represented, for example, by a recurrent network.

Since MAML has the same representational power as any other universal meta-learner, the next question we might ask is: what is the benefit of using MAML over any other approach? We study this question by analyzing the effect of continuing optimization on MAML performance. Although MAML optimizes a network's parameters for maximal performance after a fixed small number of gradient steps, we analyze the effect of taking substantially more gradient steps at meta-test time. We find that initializations learned by MAML are extremely resilient to overfitting to tiny datasets, in stark contrast to more conventional network initialization, even when taking many more gradient steps than were used during meta-training. We also find that the MAML initialization is substantially better suited for extrapolation beyond the distribution of tasks seen at meta-training time, when compared to meta-learning methods based on networks that ingest the entire training set. We analyze this setting empirically and provide some intuition to explain this effect.

## 2 PRELIMINARIES

In this section, we review the universal function approximation theorem and its extensions that we will use when considering the universal approximation of learning algorithms. We also overview the model-agnostic meta-learning algorithm and an architectural extension that we will use in Section 4.

### 2.1 UNIVERSAL FUNCTION APPROXIMATION

The universal function approximation theorem states that a neural network with one hidden layer of finite width can approximate any continuous function on compact subsets of $\mathbb{R}^n$ up to arbitrary precision (Hornik et al., 1989; Cybenko, 1989; Funahashi, 1989). The theorem holds for a range of activation functions, including the sigmoid (Hornik et al., 1989) and ReLU (Sonoda & Murata, 2017) functions. A function approximator that satisfies the definition above is often referred to as a universal function approximator (UFA). Similarly, we will define a *universal learning procedure approximator* to be a UFA with input $(\mathcal{D}, \mathbf{x}^\star)$ and output $\mathbf{y}^\star$, where $(\mathcal{D}, \mathbf{x}^\star)$ denotes the training dataset and test input, while $\mathbf{y}^\star$ denotes the desired test output. Furthermore, Hornik et al. (1990) showed that a neural network with a single hidden layer can simultaneously approximate any function and its derivatives, under mild assumptions on the activation function used and target function's domain. We will use this property in Section 4 as part of our meta-learning universality result.

### 2.2 MODEL-AGNOSTIC META-LEARNING WITH A BIAS TRANSFORMATION

Model-Agnostic Meta-Learning (MAML) is a method that proposes to learn an initial set of parameters $\theta$ such that one or a few gradient steps on $\theta$ computed using a small amount of data for one task leads to effective generalization on that task (Finn et al., 2017a). Tasks typically correspond to supervised classification or regression problems, but can also correspond to reinforcement learning problems. The MAML objective is computed over many tasks $\{\mathcal{T}_j\}$ as follows:

$$\min_\theta \sum_j \mathcal{L}(\mathcal{D}'_{\mathcal{T}_j}, \theta'_{\mathcal{T}_j}) = \sum_j \mathcal{L}(\mathcal{D}'_{\mathcal{T}_j}, \theta - \alpha \nabla_\theta \mathcal{L}(\mathcal{D}_{\mathcal{T}_j}, \theta)),$$

where $\mathcal{D}_{\mathcal{T}_j}$ corresponds to a training set for task $\mathcal{T}_j$ and the outer loss evaluates generalization on test data in $\mathcal{D}'_{\mathcal{T}_j}$. The inner optimization to compute $\theta'_{\mathcal{T}_j}$ can use multiple gradient steps; though, in this paper, we will focus on the single gradient step setting. After meta-training on a wide range of tasks, the model can quickly and efficiently learn new, held-out test tasks by running gradient descent starting from the meta-learned representation $\theta$.

While MAML is compatible with any neural network architecture and any differentiable loss function, recent work has observed that some architectural choices can improve its performance. A particularly effective modification, introduced by Finn et al. (2017b), is to concatenate a vector of

parameters, $\theta_b$, to the input. As with all other model parameters, $\theta_b$ is updated in the inner loop via gradient descent, and the initial value of $\theta_b$ is meta-learned. This modification, referred to as a bias transformation, increases the expressive power of the error gradient without changing the expressivity of the model itself. While Finn et al. (2017b) report empirical benefit from this modification, we will use this architectural design as a symmetry-breaking mechanism in our universality proof.

## 3 META-LEARNING AND UNIVERSALITY

We can broadly classify RNN-based meta-learning methods into two categories. In the first approach (Santoro et al., 2016; Duan et al., 2016; Wang et al., 2016; Mishra et al., 2018), there is a meta-learner model $g$ with parameters $\phi$ which takes as input the dataset $\mathcal{D}_\mathcal{T}$ for a particular task $\mathcal{T}$ and a new test input $\mathbf{x}^\star$, and outputs the estimated output $\hat{\mathbf{y}}^\star$ for that input:

$$\hat{\mathbf{y}}^\star = g(\mathcal{D}_\mathcal{T}, \mathbf{x}^\star; \phi) = g((\mathbf{x}, \mathbf{y})_1, ..., (\mathbf{x}, \mathbf{y})_K, \mathbf{x}^\star; \phi)$$

The meta-learner $g$ is typically a recurrent model that iterates over the dataset $\mathcal{D}$ and the new input $\mathbf{x}^\star$. For a recurrent neural network model that satisfies the UFA theorem, this approach is maximally expressive, as it can represent any function on the dataset $\mathcal{D}_\mathcal{T}$ and test input $\mathbf{x}^\star$.

In the second approach (Hochreiter et al., 2001; Bengio et al., 1992; Li & Malik, 2017b; Andrychowicz et al., 2016; Ravi & Larochelle, 2017; Ha et al., 2017), there is a meta-learner $g$ that takes as input the dataset for a particular task $\mathcal{D}_\mathcal{T}$ and the current weights $\theta$ of a learner model $f$, and outputs new parameters $\theta'_\mathcal{T}$ for the learner model. Then, the test input $\mathbf{x}^\star$ is fed into the learner model to produce the predicted output $\hat{\mathbf{y}}^\star$. The process can be written as follows:

$$\hat{\mathbf{y}}^\star = f(\mathbf{x}^\star; \theta'_\mathcal{T}) = f(\mathbf{x}^\star; g(\mathcal{D}_\mathcal{T}; \phi)) = f(\mathbf{x}^\star; g((\mathbf{x}, \mathbf{y})_{1:K}; \phi))$$

Note that, in the form written above, this approach can be as expressive as the previous approach, since the meta-learner could simply copy the dataset into some of the predicted weights, reducing to a model that takes as input the dataset and the test example.[1] Several versions of this approach, i.e. Ravi & Larochelle (2017); Li & Malik (2017b), have the recurrent meta-learner operate on order-invariant features such as the gradient and objective value averaged over the datapoints in the dataset, rather than operating on the individual datapoints themselves. This induces a potentially helpful inductive bias that disallows coupling between datapoints, ignoring the ordering within the dataset. As a result, the meta-learning process can only produce permutation-invariant functions of the dataset.

In model-agnostic meta-learning (MAML), instead of using an RNN to update the weights of the learner $f$, standard gradient descent is used. Specifically, the prediction $\hat{\mathbf{y}}^\star$ for a test input $\mathbf{x}^\star$ is:

$$\hat{\mathbf{y}}^\star = f_{\text{MAML}}(\mathcal{D}_\mathcal{T}, \mathbf{x}^\star; \theta)$$

$$= f(\mathbf{x}^\star; \theta'_\mathcal{T}) = f(\mathbf{x}^\star; \theta - \alpha \nabla_\theta \mathcal{L}(\mathcal{D}_\mathcal{T}, \theta)) = f\left(\mathbf{x}^\star; \theta - \alpha \nabla_\theta \frac{1}{K} \sum_{k=1}^{K} \ell(\mathbf{y}_k, f(\mathbf{x}_k; \theta))\right),$$

where $\theta$ denotes the initial parameters of the model $f$ and also corresponds to the parameters that are meta-learned, and $\ell$ corresponds to a loss function with respect to the label and prediction. Since the RNN approaches can approximate any update rule, they are clearly at least as expressive as gradient descent. It is less obvious whether or not the MAML update imposes any constraints on the learning procedures that can be acquired. To study this question, we define a *universal learning procedure approximator* to be a learner which can approximate any function of the set of training datapoints $\mathcal{D}_\mathcal{T}$ and the test point $\mathbf{x}^\star$. It is clear how $f_{\text{MAML}}$ can approximate any function on $\mathbf{x}^\star$, as per the UFA theorem; however, it is not obvious if $f_{\text{MAML}}$ can represent any function of the set of input, output pairs in $\mathcal{D}_\mathcal{T}$, since the UFA theorem does not consider the gradient operator.

The first goal of this paper is to show that $f_{\text{MAML}}(\mathcal{D}_\mathcal{T}, \mathbf{x}^\star; \theta)$ is a universal function approximator of $(\mathcal{D}_\mathcal{T}, \mathbf{x}^\star)$ in the one-shot setting, where the dataset $\mathcal{D}_\mathcal{T}$ consists of a single datapoint $(\mathbf{x}, \mathbf{y})$. Then, we will consider the case of $K$-shot learning, showing that $f_{\text{MAML}}(\mathcal{D}_\mathcal{T}, \mathbf{x}^\star; \theta)$ is universal in the set of functions that are invariant to the permutation of datapoints. In both cases, we will discuss meta supervised learning problems with both discrete and continuous labels and the loss functions under which universality does or does not hold.

---

[1] For this to be possible, the model $f$ must be a neural network with at least two hidden layers, since the dataset can be copied into the first layer of weights and the predicted output must be a universal function approximator of both the dataset and the test input.

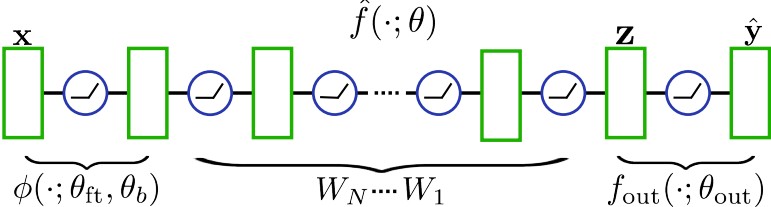

Figure 1: A deep fully-connected neural network with N+2 layers and ReLU nonlinearities. With this generic fully connected network, we prove that, with a single step of gradient descent, the model can approximate any function of the dataset and test input.

## 4 UNIVERSALITY OF THE ONE-SHOT GRADIENT-BASED LEARNER

We first introduce a proof of the universality of gradient-based meta-learning for the special case with only one training point, corresponding to one-shot learning. We denote the training datapoint as $(\mathbf{x}, \mathbf{y})$, and the test input as $\mathbf{x}^\star$. A universal learning algorithm approximator corresponds to the ability of a meta-learner to represent any function $f_{\text{target}}(\mathbf{x}, \mathbf{y}, \mathbf{x}^\star)$ up to arbitrary precision.

We will proceed by construction, showing that there exists a neural network function $\hat{f}(\cdot; \theta)$ such that $\hat{f}(\mathbf{x}^\star; \theta')$ approximates $f_{\text{target}}(\mathbf{x}, \mathbf{y}, \mathbf{x}^\star)$ up to arbitrary precision, where $\theta' = \theta - \alpha \nabla_\theta \ell(\mathbf{y}, f(\mathbf{x}))$ and $\alpha$ is the non-zero learning rate. The proof holds for a standard multi-layer ReLU network, provided that it has sufficient depth. As we discuss in Section 6, the loss function $\ell$ cannot be any loss function, but the standard cross-entropy and mean-squared error objectives are both suitable. In this proof, we will start by presenting the form of $\hat{f}$ and deriving its value after one gradient step. Then, to show universality, we will construct a setting of the weight matrices that enables independent control of the information flow coming forward from $\mathbf{x}$ and $\mathbf{x}^\star$, and backward from $\mathbf{y}$.

We will start by constructing $\hat{f}$, which, as shown in Figure 1 is a generic deep network with $N + 2$ layers and ReLU nonlinearities. Note that, for a particular weight matrix $W_i$ at layer $i$, a single gradient step $W_i - \alpha \nabla_{W_i} \ell$ can only represent a rank-1 update to the matrix $W_i$. That is because the gradient of $W_i$ is the outer product of two vectors, $\nabla_{W_i} \ell = \mathbf{a}_i \mathbf{b}_{i-1}^T$, where $\mathbf{a}_i$ is the error gradient with respect to the pre-synaptic activations at layer $i$, and $\mathbf{b}_{i-1}$ is the forward post-synaptic activations at layer $i - 1$. The expressive power of a single gradient update to a single weight matrix is therefore quite limited. However, if we sequence $N$ weight matrices as $\prod_{i=1}^{N} W_i$, corresponding to multiple linear layers, it is possible to acquire a rank-$N$ update to the linear function represented by $W = \prod_{i=1}^{N} W_i$. Note that deep ReLU networks act like deep linear networks when the input and pre-synaptic activations are non-negative. Motivated by this reasoning, we will construct $\hat{f}(\cdot; \theta)$ as a deep ReLU network where a number of the intermediate layers act as linear layers, which we ensure by showing that the input and pre-synaptic activations of these layers are non-negative. This allows us to simplify the analysis. The simplified form of the model is as follows:

$$\hat{f}(\cdot; \theta) = f_{\text{out}}\left(\left(\prod_{i=1}^{N} W_i\right) \phi(\cdot; \theta_{\text{ft}}, \theta_b); \theta_{\text{out}}\right),$$

where $\phi(\cdot; \theta_{\text{ft}}, \theta_b)$ represents an input feature extractor with parameters $\theta_{\text{ft}}$ and a scalar bias transformation variable $\theta_b$, $\prod_{i=1}^{N} W_i$ is a product of square linear weight matrices, $f_{\text{out}}(\cdot, \theta_{\text{out}})$ is a function at the output, and the learned parameters are $\theta := \{\theta_{\text{ft}}, \theta_b, \{W_i\}, \theta_{\text{out}}\}$. The input feature extractor and output function can be represented with fully connected neural networks with one or more hidden layers, which we know are universal function approximators, while $\prod_{i=1}^{N} W_i$ corresponds to a set of linear layers with non-negative input and activations.

Next, we derive the form of the post-update prediction $\hat{f}(\mathbf{x}^\star; \theta')$. Let $\mathbf{z} = \left(\prod_{i=1}^{N} W_i\right) \phi(\mathbf{x}; \theta_{\text{ft}}, \theta_b)$, and the error gradient $\nabla_{\mathbf{z}} \ell = e(\mathbf{x}, \mathbf{y})$. Then, the gradient with respect to each weight matrix $W_i$ is:

$$\nabla_{W_i} \ell(\mathbf{y}, \hat{f}(\mathbf{x}, \theta)) = \left(\prod_{j=1}^{i-1} W_j\right)^T e(\mathbf{x}, \mathbf{y}) \phi(\mathbf{x}; \theta_{\text{ft}}, \theta_b)^T \left(\prod_{j=i+1}^{N} W_j\right)^T.$$

Therefore, the post-update value of $\prod_{i=1}^{N} W_i' = \prod_{i=1}^{N}(W_i - \alpha \nabla_{W_i} \ell)$ is given by

$$\prod_{i=1}^{N} W_i - \alpha \sum_{i=1}^{N} \left(\prod_{j=1}^{i-1} W_j\right) \left(\prod_{j=1}^{i-1} W_j\right)^T e(\mathbf{x}, \mathbf{y}) \phi(\mathbf{x}; \theta_{\text{ft}}, \theta_b)^T \left(\prod_{j=i+1}^{N} W_j\right)^T \left(\prod_{j=i+1}^{N} W_j\right) - O(\alpha^2),$$

where we will disregard the last term, assuming that $\alpha$ is comparatively small such that $\alpha^2$ and all higher order terms vanish. In general, these terms do not necessarily need to vanish, and likely would further improve the expressiveness of the gradient update, but we disregard them here for the sake of the simplicity of the derivation. Ignoring these terms, we now note that the post-update value of $\mathbf{z}^\star$ when $\mathbf{x}^\star$ is provided as input into $\hat{f}(\cdot; \theta')$ is given by

$$\mathbf{z}^\star = \prod_{i=1}^{N} W_i \phi(\mathbf{x}^\star; \theta'_{\text{ft}}, \theta'_b) \tag{1}$$

$$-\alpha \sum_{i=1}^{N} \left( \prod_{j=1}^{i-1} W_j \right) \left( \prod_{j=1}^{i-1} W_j \right)^T e(\mathbf{x}, \mathbf{y}) \phi(\mathbf{x}; \theta_{\text{ft}}, \theta_b)^T \left( \prod_{j=i+1}^{N} W_j \right)^T \left( \prod_{j=i+1}^{N} W_j \right) \phi(\mathbf{x}^\star; \theta'_{\text{ft}}, \theta'_b),$$

and $\hat{f}(\mathbf{x}^\star; \theta') = f_{\text{out}}(\mathbf{z}^\star; \theta'_{\text{out}})$.

Our goal is to show that that there exists a setting of $W_i$, $f_{\text{out}}$, and $\phi$ for which the above function, $\hat{f}(\mathbf{x}^\star, \theta')$, can approximate any function of $(\mathbf{x}, \mathbf{y}, \mathbf{x}^\star)$. To show universality, we will aim to independently control information flow from $\mathbf{x}$, from $\mathbf{y}$, and from $\mathbf{x}^\star$ by multiplexing forward information from $\mathbf{x}$ and backward information from $\mathbf{y}$. We will achieve this by decomposing $W_i$, $\phi$, and the error gradient into three parts, as follows:

$$W_i := \begin{bmatrix} \tilde{W}_i & 0 & 0 \\ 0 & \overline{W}_i & 0 \\ 0 & 0 & \check{w}_i \end{bmatrix} \qquad \phi(\cdot; \theta_{\text{ft}}, \theta_b) := \begin{bmatrix} \tilde{\phi}(\cdot; \theta_{\text{ft}}, \theta_b) \\ \mathbf{0} \\ \theta_b \end{bmatrix} \qquad \nabla_{\mathbf{z}}\ell(\mathbf{y}, \hat{f}(\mathbf{x}; \theta)) := \begin{bmatrix} \mathbf{0} \\ \overline{e}(\mathbf{y}) \\ \check{e}(\mathbf{y}) \end{bmatrix} \tag{2}$$

where the initial value of $\theta_b$ will be 0. The top components all have equal numbers of rows, as do the middle components. As a result, we can see that $\mathbf{z}$ will likewise be made up of three components, which we will denote as $\tilde{\mathbf{z}}$, $\overline{\mathbf{z}}$, and $\check{z}$. Lastly, we construct the top component of the error gradient to be $\mathbf{0}$, whereas the middle and bottom components, $\overline{e}(\mathbf{y})$ and $\check{e}(\mathbf{y})$, can be set to be any linear (but not affine) function of $\mathbf{y}$. We will discuss how to achieve this gradient in the latter part of this section when we define $f_{\text{out}}$ and in Section 6.

In Appendix A.3, we show that we can choose a particular form of $\tilde{W}_i$, $\overline{W}_i$, and $\check{w}_i$ that will simplify the products of $W_j$ matrices in Equation 1, such that we get the following form for $\overline{\mathbf{z}}^\star$:

$$\overline{\mathbf{z}}^\star = -\alpha \sum_{i=1}^{N} A_i \overline{e}(\mathbf{y}) \tilde{\phi}(\mathbf{x}; \theta_{\text{ft}}, \theta_b)^T B_i^T B_i \tilde{\phi}(\mathbf{x}^\star; \theta_{\text{ft}}, \theta'_b), \tag{3}$$

where $A_1 = I$, $B_N = I$, $A_i$ can be chosen to be any symmetric positive-definite matrix, and $B_i$ can be chosen to be any positive definite matrix. In Appendix D, we further show that these definitions of the weight matrices satisfy the condition that the activations are non-negative, meaning that the model $\hat{f}$ can be represented by a generic deep network with ReLU nonlinearities.

Finally, we need to define the function $f_{\text{out}}$ at the output. When the training input $\mathbf{x}$ is passed in, we need $f_{\text{out}}$ to propagate information about the label $\mathbf{y}$ as defined in Equation 2. And, when the test input $\mathbf{x}^\star$ is passed in, we need a different function defined only on $\overline{\mathbf{z}}^\star$. Thus, we will define $f_{\text{out}}$ as a neural network that approximates the following multiplexer function and its derivatives (as shown possible by Hornik et al. (1990)):

$$f_{\text{out}}\left( \begin{bmatrix} \tilde{\mathbf{z}} \\ \overline{\mathbf{z}} \\ \check{z} \end{bmatrix}; \theta_{\text{out}} \right) = \mathbb{1}(\overline{\mathbf{z}} = \mathbf{0}) g_{\text{pre}}\left( \begin{bmatrix} \tilde{\mathbf{z}} \\ \overline{\mathbf{z}} \\ \check{z} \end{bmatrix}; \theta_g \right) + \mathbb{1}(\overline{\mathbf{z}} \neq \mathbf{0}) h_{\text{post}}(\overline{\mathbf{z}}; \theta_h), \tag{4}$$

where $g_{\text{pre}}$ is a linear function with parameters $\theta_g$ such that $\nabla_{\mathbf{z}}\ell = e(\mathbf{y})$ satisfies Equation 2 (see Section 6) and $h_{\text{post}}(\cdot; \theta_h)$ is a neural network with one or more hidden layers. As shown in Appendix A.4, the post-update value of $f_{\text{out}}$ is

$$f_{\text{out}}\left( \begin{bmatrix} \tilde{\mathbf{z}}^\star \\ \overline{\mathbf{z}}^\star \\ \check{z}^\star \end{bmatrix}; \theta'_{\text{out}} \right) = h_{\text{post}}(\overline{\mathbf{z}}^\star; \theta_h). \tag{5}$$

Now, combining Equations 3 and 5, we can see that the post-update value is the following:

$$\hat{f}(\mathbf{x}^\star; \theta') = h_{\text{post}}\left( -\alpha \sum_{i=1}^{N} A_i \overline{e}(\mathbf{y}) \tilde{\phi}(\mathbf{x}; \theta_{\text{ft}}, \theta_b)^T B_i^T B_i \tilde{\phi}(\mathbf{x}^\star; \theta_{\text{ft}}, \theta'_b); \theta_h \right) \tag{6}$$

In summary, so far, we have chosen a particular form of weight matrices, feature extractor, and output function to decouple forward and backward information flow and recover the post-update function above. Now, our goal is to show that the above function $\hat{f}(\mathbf{x}^\star; \theta')$ is a universal learning algorithm approximator, as a function of $(\mathbf{x}, \mathbf{y}, \mathbf{x}^\star)$. For notational clarity, we will use $k_i(\mathbf{x}, \mathbf{x}^\star) := \tilde{\phi}(\mathbf{x}; \theta_{\text{ft}}, \theta_b)^T B_i^T B_i \tilde{\phi}(\mathbf{x}^\star; \theta_{\text{ft}}, \theta_b')$ to denote the inner product in the above equation, noting that it can be viewed as a type of kernel with the RKHS defined by $B_i \tilde{\phi}(\mathbf{x}; \theta_{\text{ft}}, \theta_b)$.[2] The connection to kernels is not in fact needed for the proof, but provides for convenient notation and an interesting observation. We then define the following lemma:

**Lemma 4.1** *Let us assume that $\overline{e}(\mathbf{y})$ can be chosen to be any linear (but not affine) function of $\mathbf{y}$. Then, we can choose $\theta_{ft}$, $\theta_h$, $\{A_i; i > 1\}$, $\{B_i; i < N\}$ such that the function*

$$\hat{f}(\mathbf{x}^\star; \theta') = h_{post}\left(-\alpha \sum_{i=1}^{N} A_i \overline{e}(\mathbf{y}) k_i(\mathbf{x}, \mathbf{x}^\star); \theta_h\right) \tag{7}$$

*can approximate any continuous function of $(\mathbf{x}, \mathbf{y}, \mathbf{x}^\star)$ on compact subsets of $\mathbb{R}^{\dim(\mathbf{y})}$.[3]*

Intuitively, Equation 7 can be viewed as a sum of basis vectors $A_i \overline{e}(\mathbf{y})$ weighted by $k_i(\mathbf{x}, \mathbf{x}^\star)$, which is passed into $h_{\text{post}}$ to produce the output. There are likely a number of ways to prove Lemma 4.1. In Appendix A.1, we provide a simple though inefficient proof, which we will briefly summarize here. We can define $k_i$ to be a indicator function, indicating when $(\mathbf{x}, \mathbf{x}^\star)$ takes on a particular value indexed by $i$. Then, we can define $A_i \overline{e}(\mathbf{y})$ to be a vector containing the information of $\mathbf{y}$ and $i$. Then, the result of the summation will be a vector containing information about the label $\mathbf{y}$ and the value of $(\mathbf{x}, \mathbf{x}^\star)$ which is indexed by $i$. Finally, $h_{\text{post}}$ defines the output for each value of $(\mathbf{x}, \mathbf{y}, \mathbf{x}^\star)$. The bias transformation variable $\theta_b$ plays a vital role in our construction, as it breaks the symmetry within $k_i(\mathbf{x}, \mathbf{x}^\star)$. Without such asymmetry, it would not be possible for our constructed function to represent any function of $\mathbf{x}$ and $\mathbf{x}^\star$ after one gradient step.

In conclusion, we have shown that there exists a neural network structure for which $\hat{f}(\mathbf{x}^\star; \theta')$ is a universal approximator of $f_{\text{target}}(\mathbf{x}, \mathbf{y}, \mathbf{x}^\star)$. We chose a particular form of $\hat{f}(\cdot; \theta)$ that decouples forward and backward information flow. With this choice, it is possible to impose any desired post-update function, even in the face of adversarial training datasets and loss functions, e.g. when the gradient points in the wrong direction. If we make the assumption that the inner loss function and training dataset are not chosen adversarially and the error gradient points in the direction of improvement, it is likely that a much simpler architecture will suffice that does not require multiplexing of forward and backward information in separate channels. Informative loss functions and training data allowing for simpler functions is indicative of the inductive bias built into gradient-based meta-learners, which is not present in recurrent meta-learners.

Our result in this section implies that a sufficiently deep representation combined with just a single gradient step can approximate any one-shot learning algorithm. In the next section, we will show the universality of MAML for $K$-shot learning algorithms.

## 5 GENERAL UNIVERSALITY OF THE GRADIENT-BASED LEARNER

Now, we consider the more general $K$-shot setting, aiming to show that MAML can approximate any permutation invariant function of a dataset and test datapoint $(\{(\mathbf{x}, \mathbf{y})_i; i \in 1...K\}, \mathbf{x}^\star)$ for $K > 1$. Note that $K$ does not need to be small. To reduce redundancy, we will only overview the differences from the 1-shot setting in this section. We include a full proof in Appendix B.

In the $K$-shot setting, the parameters of $\hat{f}(\cdot, \theta)$ are updated according to the following rule:

$$\theta' = \theta - \alpha \frac{1}{K} \sum_{k=1}^{K} \nabla_\theta \ell(\mathbf{y}_k, f(\mathbf{x}_k; \theta))).$$

---

[2]Due to the symmetry of kernels, this requires interpreting $\theta_b$ as part of the input, rather than a kernel hyperparameter, so that the left input is $(\mathbf{x}, \theta_b)$ and the right one is $(\mathbf{x}^\star, \theta_b')$.

[3]The assumption with regard to compact subsets of the output space is inherited from the UFA theorem.

Defining the form of $\hat{f}$ to be the same as in Section 4, the post-update function is the following:

$$\hat{f}(\mathbf{x}^\star; \theta') = h_{\text{post}}\left(-\alpha\frac{1}{K}\sum_{i=1}^{N}\sum_{k=1}^{K} A_i \overline{e}(\mathbf{y}_k)k_i(\mathbf{x}_k, \mathbf{x}^\star); \theta_h\right)$$

In Appendix C, we show one way in which this function can approximate any function of $(\{(\mathbf{x}, \mathbf{y})_k; k \in 1...K\}, \mathbf{x}^\star)$ that is invariant to the ordering of the training datapoints $\{(\mathbf{x}, \mathbf{y})_k; k \in 1...K\}$. We do so by showing that we can select a setting of $\tilde{\phi}$ and of each $A_i$ and $B_i$ such that $\overline{\mathbf{z}}^\star$ is a vector containing a discretization of $\mathbf{x}^\star$ and frequency counts of the discretized datapoints[4]. If $\overline{\mathbf{z}}^\star$ is a vector that completely describes $(\{(\mathbf{x}, \mathbf{y})_i\}, \mathbf{x}^\star)$ without loss of information and because $h_{\text{post}}$ is a universal function approximator, $\hat{f}(\mathbf{x}^\star; \theta')$ can approximate any continuous function of $(\{(\mathbf{x}, \mathbf{y})_i\}, \mathbf{x}^\star)$ on compact subsets of $\mathbb{R}^{\dim(\mathbf{y})}$. It's also worth noting that the form of the above equation greatly resembles a kernel-based function approximator around the training points, and a substantially more efficient universality proof can likely be obtained starting from this premise.

## 6 LOSS FUNCTIONS

In the previous sections, we showed that a deep representation combined with gradient descent can approximate any learning algorithm. In this section, we will discuss the requirements that the loss function must satisfy in order for the results in Sections 4 and 5 to hold. As one might expect, the main requirement will be for the label to be recoverable from the gradient of the loss.

As seen in the definition of $f_{\text{out}}$ in Equation 4, the pre-update function $\hat{f}(\mathbf{x}, \theta)$ is given by $g_{\text{pre}}(\mathbf{z}; \theta_g)$, where $g_{\text{pre}}$ is used for back-propagating information about the label(s) to the learner. As stated in Equation 2, we require that the error gradient with respect to $\mathbf{z}$ to be:

$$\nabla_{\mathbf{z}}\ell(\mathbf{y}, \hat{f}(\mathbf{x}; \theta)) = \begin{bmatrix} \mathbf{0} \\ \overline{e}(\mathbf{y}) \\ \check{e}(\mathbf{y}) \end{bmatrix}, \quad \text{where } \mathbf{z} = \begin{bmatrix} \tilde{\mathbf{z}} \\ \overline{\mathbf{z}} \\ \theta_b \end{bmatrix} = \begin{bmatrix} \tilde{\phi}(\mathbf{x}; \theta_{\text{ft}}, \theta_b) \\ \mathbf{0} \\ 0 \end{bmatrix},$$

and where $\overline{e}(\mathbf{y})$ and $\check{e}(\mathbf{y})$ must be able to represent [at least] any linear function of the label $\mathbf{y}$.

We define $g_{\text{pre}}$ as follows: $g_{\text{pre}}(\mathbf{z}) := \begin{bmatrix} \tilde{W}_g & \overline{W}_g & \check{\mathbf{w}}_g \end{bmatrix} \mathbf{z} = \tilde{W}_g\tilde{\mathbf{z}} + \overline{W}_g\overline{\mathbf{z}} + \theta_b\check{\mathbf{w}}_g$.

To make the top term of the gradient equal to $\mathbf{0}$, we can set $\tilde{W}_g$ to be 0, which causes the pre-update prediction $\hat{\mathbf{y}} = \hat{f}(\mathbf{x}, \theta)$ to be $\mathbf{0}$. Next, note that $\overline{e}(\mathbf{y}) = \overline{W}_g^T\nabla_{\hat{\mathbf{y}}}\ell(\mathbf{y}, \hat{\mathbf{y}})$ and $\check{e}(\mathbf{y}) = \check{\mathbf{w}}_g^T\nabla_{\hat{\mathbf{y}}}\ell(\mathbf{y}, \hat{\mathbf{y}})$. Thus, for $e(\mathbf{y})$ to be any linear function of $\mathbf{y}$, we require a loss function for which $\nabla_{\hat{\mathbf{y}}}\ell(\mathbf{y}, \mathbf{0})$ is a linear function $A\mathbf{y}$, where $A$ is invertible. Essentially, $\mathbf{y}$ needs to be recoverable from the loss function's gradient. In Appendix E and F, we prove the following two theorems, thus showing that the standard $\ell_2$ and cross-entropy losses allow for the universality of gradient-based meta-learning.

**Theorem 6.1** *The gradient of the standard mean-squared error objective evaluated at $\hat{\mathbf{y}} = \mathbf{0}$ is a linear, invertible function of $\mathbf{y}$.*

**Theorem 6.2** *The gradient of the softmax cross entropy loss with respect to the pre-softmax logits is a linear, invertible function of $\mathbf{y}$, when evaluated at $\mathbf{0}$.*

Now consider other popular loss functions whose gradients do not satisfy the label-linearity property. The gradients of the $\ell_1$ and hinge losses are piecewise constant, and thus do not allow for universality. The Huber loss is also piecewise constant in some areas its domain. These error functions effectively lose information because simply looking at their gradient is insufficient to determine the label. Recurrent meta-learners that take the gradient as input, rather than the label, e.g. Andrychowicz et al. (2016), will also suffer from this loss of information when using these error functions.

## 7 EXPERIMENTS

Now that we have shown that meta-learners that use standard gradient descent with a sufficiently deep representation can approximate any learning procedure, and are equally expressive as recurrent learners, a natural next question is – is there empirical benefit to using one meta-learning approach versus another, and in which cases? To answer this question, we next aim to empirically study

---

[4]With continuous labels $\mathbf{y}$ and mean-squared error $\ell$, we require the mild assumption that no two datapoints may share the same input value $\mathbf{x}$: the input datapoints must be unique.

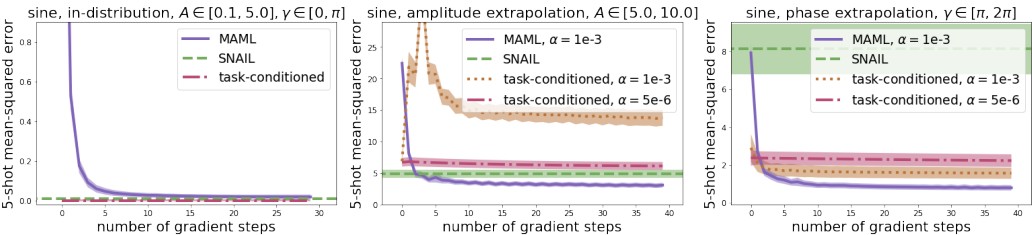

Figure 2: The effect of additional gradient steps at test time when attempting to solve new tasks. The MAML model, trained with 5 inner gradient steps, can further improve with more steps. All methods are provided with the same data – 5 examples – where each gradient step is computed using the same 5 datapoints.

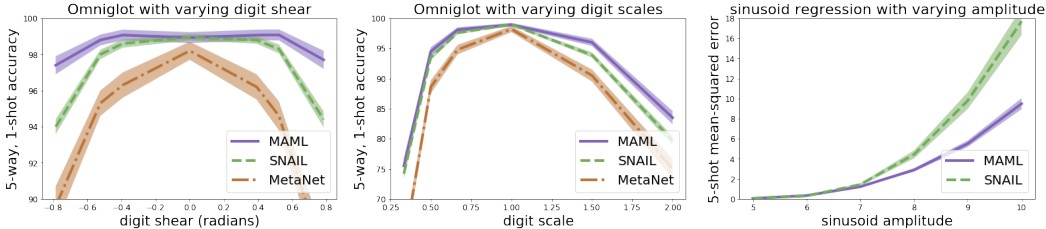

Figure 3: Learning performance on out-of-distribution tasks as a function of the task variability. Recurrent meta-learners such as SNAIL and MetaNet acquire learning strategies that are less generalizable than those learned with gradient-based meta-learning.

the inductive bias of gradient-based and recurrent meta-learners. Then, in Section 7.2, we will investigate the role of model depth in gradient-based meta-learning, as the theory suggests that deeper networks lead to increased expressive power for representing different learning procedures.

## 7.1 EMPIRICAL STUDY OF INDUCTIVE BIAS

First, we aim to empirically explore the differences between gradient-based and recurrent meta-learners. In particular, we aim to answer the following questions: (1) can a learner trained with MAML further improve from additional gradient steps when learning new tasks at test time, or does it start to overfit? and (2) does the inductive bias of gradient descent enable better few-shot learning performance on tasks outside of the training distribution, compared to learning algorithms represented as recurrent networks?

To study both questions, we will consider two simple few-shot learning domains. The first is 5-shot regression on a family of sine curves with varying amplitude and phase. We trained all models on a uniform distribution of tasks with amplitudes $A \in [0.1, 5.0]$, and phases $\gamma \in [0, \pi]$. The second domain is 1-shot character classification using the Omniglot dataset (Lake et al., 2011), following the training protocol introduced by Santoro et al. (2016). In our comparisons to recurrent meta-learners, we will use two state-of-the-art meta-learning models: SNAIL (Mishra et al., 2018) and meta-networks (Munkhdalai & Yu, 2017). In some experiments, we will also compare to a task-conditioned model, which is trained to map from both the input and the task description to the label. Like MAML, the task-conditioned model can be fine-tuned on new data using gradient descent, but is not trained for few-shot adaptation. We include more experimental details in Appendix G.

To answer the first question, we fine-tuned a model trained using MAML with many more gradient steps than used during

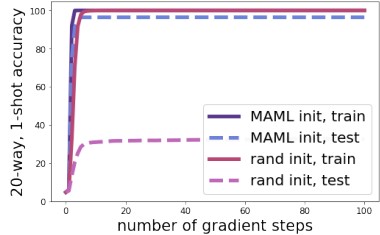

Figure 4: Comparison of finetuning from a MAML-initialized network and a network initialized randomly, trained from scratch. Both methods achieve about the same training accuracy. But, MAML also attains good test accuracy, while the network trained from scratch overfits catastrophically to the 20 examples. Interestingly, the MAML-initialized model does not begin to overfit, even though meta-training used 5 steps while the graph shows up to 100.

meta-training. The results on the sinusoid domain, shown in Figure 2, show that a MAML-learned initialization trained for fast adaption in 5 steps can further improve beyond 5 gradient steps, especially on out-of-distribution tasks. In contrast, a task-conditioned model trained without MAML can easily overfit to out-of-distribution tasks. With the Omniglot dataset, as seen in Figure 4, a MAML model that was trained with 5 inner gradient steps can be fine-tuned for 100 gradient steps without leading to any drop in test accuracy. As expected, a model initialized randomly and trained from scratch quickly reaches perfect training accuracy, but overfits massively to the 20 examples.

Next, we investigate the second question, aiming to compare MAML with state-of-the-art recurrent meta-learners on tasks that are related to, but outside of the distribution of the training tasks. All three methods achieved similar performance within the distribution of training tasks for 5-way 1-shot Omniglot classification and 5-shot sinusoid regression. In the Omniglot setting, we compare each method's ability to distinguish digits that have been sheared or scaled by varying amounts. In the sinusoid regression setting, we compare on sinusoids with extrapolated amplitudes within $[5.0, 10.0]$ and phases within $[\pi, 2\pi]$. The results in Figure 3 and Appendix G show a clear trend that MAML recovers more generalizable learning strategies. Combined with the theoretical universality results, these experiments indicate that deep gradient-based meta-learners are not only equivalent in representational power to recurrent meta-learners, but should also be a considered as a strong contender in settings that contain domain shift between meta-training and meta-testing tasks, where their strong inductive bias for reasonable learning strategies provides substantially improved performance.

## 7.2 Effect of Depth

The proofs in Sections 4 and 5 suggest that gradient descent with deeper representations results in more expressive learning procedures. In contrast, the universal function approximation theorem only requires a single hidden layer to approximate any function. Now, we seek to empirically explore this theoretical finding, aiming to answer the question: is there a scenario for which model-agnostic meta-learning requires a deeper representation to achieve good performance, compared to the depth of the representation needed to solve the underlying tasks being learned?

To answer this question, we will study a simple regression problem, where the meta-learning goal is to infer a polynomial function from 40 input/output datapoints. We use polynomials of degree 3 where the coefficients and bias are sampled uniformly at random within $[-1, 1]$ and the input values range within $[-3, 3]$. Similar to the conditions in the proof, we meta-train and

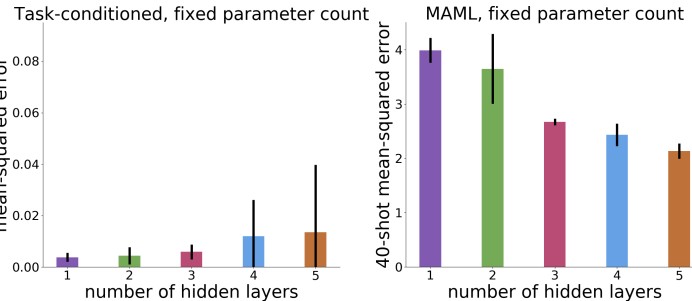

Figure 5: Comparison of depth while keeping the number of parameters constant. Task-conditioned models do not need more than one hidden layer, whereas meta-learning with MAML clearly benefits from additional depth. Error bars show standard deviation over three training runs.

meta-test with one gradient step, use a mean-squared error objective, use ReLU nonlinearities, and use a bias transformation variable of dimension 10. To compare the relationship between depth and expressive power, we will compare models with a fixed number of parameters, approximately $40,000$, and vary the network depth from 1 to 5 hidden layers. As a point of comparison to the models trained for meta-learning using MAML, we trained standard feedforward models to regress from the input and the 4-dimensional task description (the 3 coefficients of the polynomial and the scalar bias) to the output. These task-conditioned models act as an oracle and are meant to empirically determine the depth needed to represent these polynomials, independent of the meta-learning process. Theoretically, we would expect the task-conditioned models to require only one hidden layer, as per the universal function approximation theorem. In contrast, we would expect the MAML model to require more depth. The results, shown in Figure 5, demonstrate that the task-conditioned model does indeed not benefit from having more than one hidden layer, whereas the MAML clearly achieves better performance with more depth even though the model capacity, in terms of the number of parameters, is fixed. This empirical effect supports the theoretical finding that depth is important for effective meta-learning using MAML.

## 8 Conclusion

In this paper, we show that there exists a form of deep neural network such that the initial weights combined with gradient descent can approximate any learning algorithm. Our findings suggest that, from the standpoint of expressivity, there is no theoretical disadvantage to embedding gradient descent into the meta-learning process. In fact, in all of our experiments, we found that the learning strategies acquired with MAML are more successful when faced with out-of-domain tasks compared to recurrent learners. Furthermore, we show that the representations acquired with MAML are highly resilient to overfitting. These results suggest that gradient-based meta-learning has a num-

ber of practical benefits, and no theoretical downsides in terms of expressivity when compared to alternative meta-learning models. Independent of the type of meta-learning algorithm, we formalize what it means for a meta-learner to be able to approximate any learning algorithm in terms of its ability to represent functions of the dataset and test inputs. This formalism provides a new perspective on the learning-to-learn problem, which we hope will lead to further discussion and research on the goals and methodology surrounding meta-learning.

ACKNOWLEDGMENTS

We thank Sharad Vikram for detailed feedback on the proof, as well as Justin Fu, Ashvin Nair, and Kelvin Xu for feedback on an early draft of this paper. We also thank Erin Grant for helpful conversations and Nikhil Mishra for providing code for SNAIL. This research was supported by the National Science Foundation through IIS-1651843 and a Graduate Research Fellowship, as well as NVIDIA.

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

## A  SUPPLEMENTARY PROOFS FOR 1-SHOT SETTING

### A.1  PROOF OF LEMMA 4.1

While there are likely a number of ways to prove Lemma 4.1 (copied below for convenience), here we provide a simple, though inefficient, proof of Lemma 4.1.

**Lemma 4.1** *Let us assume that $\overline{e}(\mathbf{y})$ can be chosen to be any linear (but not affine) function of $\mathbf{y}$. Then, we can choose $\theta_{ft}$, $\theta_h$, $\{A_i; i > 1\}$, $\{B_i; i < N\}$ such that the function*

$$\hat{f}(\mathbf{x}^\star; \theta') = h_{post}\left(-\alpha \sum_{i=1}^{N} A_i \overline{e}(\mathbf{y}) k_i(\mathbf{x}, \mathbf{x}^\star); \theta_h\right) \tag{7}$$

*can approximate any continuous function of $(\mathbf{x}, \mathbf{y}, \mathbf{x}^\star)$ on compact subsets of $\mathbb{R}^{\dim(\mathbf{y})}$.[5]*

To prove this lemma, we will proceed by showing that we can choose $\overline{e}$, $\theta_{ft}$, and each $A_i$ and $B_i$ such that the summation contains a complete description of the values of $\mathbf{x}$, $\mathbf{x}^\star$, and $\mathbf{y}$. Then, because $h_{post}$ is a universal function approximator, $\hat{f}(\mathbf{x}^\star, \theta')$ will be able to approximate any function of $\mathbf{x}$, $\mathbf{x}^\star$, and $\mathbf{y}$.

Since $A_1 = I$ and $B_N = I$, we will essentially ignore the first and last elements of the sum by defining $B_1 := \epsilon I$ and $A_N := \epsilon I$, where $\epsilon$ is a small positive constant to ensure positive definiteness. Then, we can rewrite the summation, omitting the first and last terms:

$$\hat{f}(\mathbf{x}^\star; \theta') \approx h_{post}\left(-\alpha \sum_{i=2}^{N-1} A_i \overline{e}(\mathbf{y}) k_i(\mathbf{x}, \mathbf{x}^\star); \theta_h\right)$$

Next, we will re-index using two indexing variables, $j$ and $l$, where $j$ will index over the discretization of $\mathbf{x}$ and $l$ over the discretization of $\mathbf{x}^\star$.

$$\hat{f}(\mathbf{x}^\star; \theta') \approx h_{post}\left(-\alpha \sum_{j=0}^{J-1} \sum_{l=0}^{L-1} A_{jl} \overline{e}(\mathbf{y}) k_{jl}(\mathbf{x}, \mathbf{x}^\star); \theta_h\right)$$

Next, we will define our chosen form of $k_{jl}$ in Equation 8. We show how to acquire this form in the next section.

**Lemma A.1** *We can choose $\theta_{ft}$ and each $B_{jl}$ such that*

$$k_{jl}(\mathbf{x}, \mathbf{x}^\star) := \begin{cases} 1 & \text{if } \mathrm{discr}(\mathbf{x}) = \mathbf{e}_j \text{ and } \mathrm{discr}(\mathbf{x}^\star) = \mathbf{e}_l \\ 0 & \text{otherwise} \end{cases} \tag{8}$$

*where $\mathrm{discr}(\cdot)$ denotes a function that produces a one-hot discretization of its input and $\mathbf{e}$ denotes the 0-indexed standard basis vector.*

Now that we have defined the function $k_{jl}$, we will next define the other terms in the sum. Our goal is for the summation to contain complete information about $(\mathbf{x}, \mathbf{x}^\star, \mathbf{y})$. To do so, we will chose $\overline{e}(\mathbf{y})$ to be the linear function that outputs $J * L$ stacked copies of $\mathbf{y}$. Then, we will define $A_{jl}$ to be a matrix that selects the copy of $\mathbf{y}$ in the position corresponding to $(j, l)$, i.e. in the position $j + J * l$. This can be achieved using a diagonal $A_{jl}$ matrix with diagonal values of $1 + \epsilon$ at the positions corresponding to the $k$th vector, and $\epsilon$ elsewhere, where $k = (j + J * l)$ and $\epsilon$ is used to ensure that $A_{jl}$ is positive definite.

As a result, the post-update function is as follows:

$$\hat{f}(\mathbf{x}^\star; \theta') \approx h_{post}\left(-\alpha v(\mathbf{x}, \mathbf{x}^\star, \mathbf{y}); \theta_h\right), \text{ where } v(\mathbf{x}, \mathbf{x}^\star, \mathbf{y}) \approx \begin{bmatrix} \mathbf{0} \\ \vdots \\ \mathbf{0} \\ \mathbf{y} \\ \mathbf{0} \\ \vdots \\ \mathbf{0} \end{bmatrix},$$

---

[5]The assumption with regard to compact subsets of the output space is inherited from the UFA theorem.

where $\mathbf{y}$ is at the position $j + J * l$ within the vector $v(\mathbf{x}, \mathbf{x}^\star, \mathbf{y})$, where $j$ satisfies $\mathrm{discr}(\mathbf{x}) = \mathbf{e}_j$ and where $l$ satisfies $\mathrm{discr}(\mathbf{x}^\star) = \mathbf{e}_l$. Note that the vector $-\alpha v(\mathbf{x}, \mathbf{x}^\star, \mathbf{y})$ is a complete description of $(\mathbf{x}, \mathbf{x}^\star, \mathbf{y})$ in that $\mathbf{x}$, $\mathbf{x}^\star$, and $\mathbf{y}$ can be decoded from it. Therefore, since $h_{\mathrm{post}}$ is a universal function approximator and because its input contains all of the information of $(\mathbf{x}, \mathbf{x}^\star, \mathbf{y})$, the function $\hat{f}(\mathbf{x}^\star; \theta') \approx h_{\mathrm{post}}(-\alpha v(\mathbf{x}, \mathbf{x}^\star, \mathbf{y}); \theta_h)$ is a universal function approximator with respect to its inputs $(\mathbf{x}, \mathbf{x}^\star, \mathbf{y})$.

## A.2 PROOF OF LEMMA A.1

In this section, we show one way of proving Lemma A.1:

**Lemma A.1** *We can choose $\theta_{ft}$ and each $B_{jl}$ such that*

$$k_{jl}(\mathbf{x}, \mathbf{x}^\star) := \begin{cases} 1 & \text{if } \mathrm{discr}(\mathbf{x}) = \mathbf{e}_j \text{ and } \mathrm{discr}(\mathbf{x}^\star) = \mathbf{e}_l \\ 0 & \text{otherwise} \end{cases} \tag{8}$$

*where* $\mathrm{discr}(\cdot)$ *denotes a function that produces a one-hot discretization of its input and* $\mathbf{e}$ *denotes the 0-indexed standard basis vector.*

Recall that $k_{jl}(\mathbf{x}, \mathbf{x}^\star)$ is defined as $\tilde{\phi}(\mathbf{x}; \theta_{ft}, \theta_b)^T B_{jl}^T B_{jl} \tilde{\phi}(\mathbf{x}^\star; \theta_{ft}, \theta_b')$, where $\theta_b = 0$. Since the gradient with respect to $\theta_b$ can be chosen to be any linear function of the label $\mathbf{y}$ (see Section 6), we can assume without loss of generality that $\theta_b' \neq 0$.

We will choose $\tilde{\phi}$ and $B_{jl}$ as follows:

$$\tilde{\phi}(\cdot; \theta_{ft}, \theta_b) := \begin{cases} \begin{bmatrix} \mathrm{discr}(\cdot) \\ \mathbf{0} \end{bmatrix} & \text{if } \theta_b = 0 \\ \begin{bmatrix} \mathbf{0} \\ \mathrm{discr}(\cdot) \end{bmatrix} & \text{otherwise} \end{cases} \qquad B_{jl} = \begin{bmatrix} E_{jj} & E_{jl} \\ E_{lj} & 0 \end{bmatrix} + \epsilon I$$

where we use $E_{ik}$ to denote the matrix with a 1 at $(i, k)$ and 0 elsewhere, and $\epsilon I$ is added to ensure the positive definiteness of $B_{jl}$ as required in the construction.

Using the above definitions, we can see that:

$$\tilde{\phi}(\mathbf{x}; \theta_{ft}, 0)^T B_{jl}^T \approx \begin{cases} \begin{bmatrix} \mathbf{e}_j \\ \mathbf{0} \end{bmatrix}^T & \text{if } \mathrm{discr}(\mathbf{x}) = \mathbf{e}_j \\ \begin{bmatrix} \mathbf{0} \\ \mathbf{0} \end{bmatrix}^T & \text{otherwise} \end{cases} \qquad B_{jl}\tilde{\phi}(\mathbf{x}^\star; \theta_{ft}, \theta_b') \approx \begin{cases} \begin{bmatrix} \mathbf{e}_j \\ \mathbf{0} \end{bmatrix} & \text{if } \mathrm{discr}(\mathbf{x}^\star) = \mathbf{e}_l \\ \begin{bmatrix} \mathbf{0} \\ \mathbf{0} \end{bmatrix} & \text{otherwise} \end{cases}$$

Thus, we have proved the lemma, showing that we can choose a $\tilde{\phi}$ and each $B_{jl}$ such that:

$$k_{jl}(\mathbf{x}, \mathbf{x}^\star) \approx \begin{cases} \begin{bmatrix} \mathbf{e}_j & \mathbf{0} \end{bmatrix} \begin{bmatrix} \mathbf{e}_j \\ \mathbf{0} \end{bmatrix} = 1 & \text{if } \mathrm{discr}(\mathbf{x}) = \mathbf{e}_j \text{ and } \mathrm{discr}(\mathbf{x}^\star) = \mathbf{e}_l \\ 0 & \text{otherwise} \end{cases}$$

## A.3 FORM OF LINEAR WEIGHT MATRICES

The goal of this section is to show that we can choose a form of $\tilde{W}$, $\overline{W}$, and $\breve{w}$ such that we can simplify the form of $\mathbf{z}^\star$ in Equation 1 into the following:

$$\overline{\mathbf{z}}^\star = -\alpha \sum_{i=1}^{N} A_i \overline{e}(\mathbf{y}) \tilde{\phi}(\mathbf{x}; \theta_{ft}, \theta_b)^T B_i^T B_i \tilde{\phi}(\mathbf{x}^\star; \theta_{ft}, \theta_b'), \tag{9}$$

where $A_1 = \mathbf{I}$, $A_i = \overline{M}_{i-1} \overline{M}_{i-1}^T$ for $i > 1$, $B_i = \tilde{M}_{i+1}$ for $i < N$ and $B_N = \mathbf{I}$.

Recall that we decomposed $W_i$, $\phi$, and the error gradient into three parts, as follows:

$$W_i := \begin{bmatrix} \tilde{W}_i & 0 & 0 \\ 0 & \overline{W}_i & 0 \\ 0 & 0 & \breve{w}_i \end{bmatrix} \qquad \phi(\cdot; \theta_{ft}, \theta_b) := \begin{bmatrix} \tilde{\phi}(\cdot; \theta_{ft}, \theta_b) \\ \mathbf{0} \\ \theta_b \end{bmatrix} \qquad \nabla_{\mathbf{z}} \ell(\mathbf{y}, \hat{f}(\mathbf{x}; \theta)) := \begin{bmatrix} \mathbf{0} \\ \overline{e}(\mathbf{y}) \\ \breve{e}(\mathbf{y}) \end{bmatrix} \tag{10}$$

where the initial value of $\theta_b$ will be 0. The top components, $\tilde{W}_i$ and $\tilde{\phi}$, have equal dimensions, as do the middle components, $\overline{W}_i$ and $\mathbf{0}$. The bottom components are scalars. As a result, we can see that $\mathbf{z}$ will likewise be made up of three components, which we will denote as $\check{\mathbf{z}}$, $\overline{\mathbf{z}}$, and $\check{z}$, where, before the gradient update, $\check{\mathbf{z}} = \prod_{i=1}^{N} \tilde{W}_i \tilde{\phi}(\mathbf{x}; \theta_{\text{ft}})$, $\overline{\mathbf{z}} = \mathbf{0}$, and $\check{z} = 0$. Lastly, we construct the top component of the error gradient to be $\mathbf{0}$, whereas the middle and bottom components, $\overline{e}(\mathbf{y})$ and $\check{e}(\mathbf{y})$, can be set to be any linear (but not affine) function of $\mathbf{y}$.

Using the above definitions and noting that $\theta'_{\text{ft}} = \theta_{\text{ft}} - \alpha \nabla_{\theta_{\text{ft}}} \ell = \theta_{\text{ft}}$, we can simplify the form of $\mathbf{z}^\star$ in Equation 1, such that the middle component, $\overline{\mathbf{z}}^\star$, is the following:

$$\overline{\mathbf{z}}^\star = -\alpha \sum_{i=1}^{N} \left( \prod_{j=1}^{i-1} \overline{W}_j \right) \left( \prod_{j=1}^{i-1} \overline{W}_j \right)^T \overline{e}(\mathbf{y}) \tilde{\phi}(\mathbf{x}; \theta_{\text{ft}}, \theta_b)^T \left( \prod_{j=i+1}^{N} \tilde{W}_j \right)^T \left( \prod_{j=i+1}^{N} \tilde{W}_j \right) \tilde{\phi}(\mathbf{x}^\star; \theta_{\text{ft}}, \theta'_b)$$

We aim to independently control the backward information flow from the gradient $e$ and the forward information flow from $\tilde{\phi}$. Thus, choosing all $\tilde{W}_i$ and $\overline{W}_i$ to be square and full rank, we will set

$$\tilde{W}_i = \tilde{M}_i \tilde{M}_{i+1}^{-1} \qquad\qquad \overline{W}_i = \overline{M}_{i-1}^{-1} \overline{M}_i,$$

so that we have

$$\prod_{j=i+1}^{N} \tilde{W}_j = \tilde{M}_{i+1} \qquad\qquad \prod_{j=1}^{i-1} \overline{W}_j = \overline{M}_{i-1},$$

for $i \in \{1...N\}$ where $\tilde{M}_{N+1} = \mathbf{I}$ and $\overline{M}_0 = \mathbf{I}$. Then we can again simplify the form of $\overline{\mathbf{z}}^\star$:

$$\overline{\mathbf{z}}^\star = -\alpha \sum_{i=1}^{N} A_i \overline{e}(\mathbf{y}) \tilde{\phi}(\mathbf{x}; \theta_{\text{ft}}, \theta_b)^T B_i^T B_i \tilde{\phi}(\mathbf{x}^\star; \theta_{\text{ft}}, \theta'_b), \tag{11}$$

where $A_1 = \mathbf{I}$, $A_i = \overline{M}_{i-1} \overline{M}_{i-1}^T$ for $i > 1$, $B_i = \tilde{M}_{i+1}$ for $i < N$ and $B_N = \mathbf{I}$.

## A.4 OUTPUT FUNCTION

In this section, we will derive the post-update version of the output function $f_{\text{out}}(\cdot; \theta_{\text{out}})$. Recall that $f_{\text{out}}$ is defined as a neural network that approximates the following multiplexer function and its derivatives (as shown possible by Hornik et al. (1990)):

$$f_{\text{out}}\left( \begin{bmatrix} \tilde{\mathbf{z}} \\ \overline{\mathbf{z}} \\ \check{z} \end{bmatrix}; \theta_{\text{out}} \right) = \mathbb{1}(\overline{\mathbf{z}} = \mathbf{0}) g_{\text{pre}}\left( \begin{bmatrix} \tilde{\mathbf{z}} \\ \overline{\mathbf{z}} \\ \check{z} \end{bmatrix}; \theta_g \right) + \mathbb{1}(\overline{\mathbf{z}} \neq \mathbf{0}) h_{\text{post}}(\overline{\mathbf{z}}; \theta_h). \tag{12}$$

The parameters $\{\theta_g, \theta_h\}$ are a part of $\theta_{\text{out}}$, in addition to the parameters required to estimate the indicator functions and their corresponding products. Since $\overline{\mathbf{z}} = \mathbf{0}$ and $h_{\text{post}}(\overline{\mathbf{z}}) = \mathbf{0}$ when the gradient step is taken, we can see that the error gradients with respect to the parameters in the last term in Equation 12 will be approximately zero. Furthermore, as seen in the definition of $g_{\text{pre}}$ in Section 6, the value of $g_{\text{pre}}(\mathbf{z}, \theta_g)$ is also zero, resulting in a gradient of approximately zero for the first indicator function.[6]

The post-update value of $f_{\text{out}}$ is therefore:

$$f_{\text{out}}\left( \begin{bmatrix} \tilde{\mathbf{z}}^\star \\ \overline{\mathbf{z}}^\star \\ \check{z}^\star \end{bmatrix}; \theta'_{\text{out}} \right) \approx \mathbb{1}(\overline{\mathbf{z}}^\star = \mathbf{0}) g_{\text{pre}}\left( \begin{bmatrix} \tilde{\mathbf{z}}^\star \\ \overline{\mathbf{z}}^\star \\ \check{z}^\star \end{bmatrix}; \theta'_g \right) + \mathbb{1}(\overline{\mathbf{z}}^\star \neq \mathbf{0}) h_{\text{post}}(\overline{\mathbf{z}}^\star; \theta_h) = h_{\text{post}}(\overline{\mathbf{z}}^\star; \theta_h) \tag{13}$$

as long as $\overline{\mathbf{z}}^\star \neq \mathbf{0}$. In Appendix A.1, we can see that $\mathbf{z}^\star$ is indeed not equal to zero.

---

[6]To guarantee that g and h are zero when evaluated at $\mathbf{x}$, we make the assumption that $g_{\text{pre}}$ and $h_{\text{post}}$ are neural networks with no biases and nonlinearity functions that output zero when evaluated at zero.

# B  FULL K-SHOT PROOF OF UNIVERSALITY

In this appendix, we provide a full proof of the universality of gradient-based meta-learning in the general case with $K > 1$ datapoints. This proof will share a lot of content from the proof in the 1-shot setting, but we include it for completeness.

We aim to show that a deep representation combined with one step of gradient descent can approximate any permutation invariant function of a dataset and test datapoint $(\{(\mathbf{x}, \mathbf{y})_i; i \in 1...K\}, \mathbf{x}^\star)$ for $K > 1$. Note that $K$ does not need to be small.

We will proceed by construction, showing that there exists a neural network function $\hat{f}(\cdot; \theta)$ such that $\hat{f}(\cdot; \theta')$ approximates $f_{\text{target}}(\{(\mathbf{x}, \mathbf{y})_k\}, \mathbf{x}^\star)$ up to arbitrary precision, where $\theta' = \theta - \alpha \frac{1}{K} \sum_{k=1}^{K} \nabla_\theta \ell(\mathbf{y}_k, f(\mathbf{x}_k; \theta)))$ and $\alpha$ is the learning rate. As we discuss in Section 6, the loss function $\ell$ cannot be any loss function, but the standard cross-entropy and mean-squared error objectives are both suitable. In this proof, we will start by presenting the form of $\hat{f}$ and deriving its value after one gradient step. Then, to show universality, we will construct a setting of the weight matrices that enables independent control of the information flow coming forward from the inputs $\{\mathbf{x}_k\}$ and $\mathbf{x}^\star$, and backward from the labels $\{\mathbf{y}_k\}$.

We will start by constructing $\hat{f}$. With the same motivation as in Section 4, we will construct $\hat{f}(\cdot; \theta)$ as the following:

$$\hat{f}(\cdot; \theta) = f_{\text{out}}\left(\left(\prod_{i=1}^{N} W_i\right)\phi(\cdot; \theta_{\text{ft}}, \theta_b); \theta_{\text{out}}\right).$$

$\phi(\cdot; \theta_{\text{ft}}, \theta_b)$ represents an input feature extractor with parameters $\theta_{\text{ft}}$ and a scalar bias transformation variable $\theta_b$, $\prod_{i=1}^{N} W_i$ is a product of square linear weight matrices, $f_{\text{out}}(\cdot, \theta_{\text{out}})$ is a readout function at the output, and the learned parameters are $\theta := \{\theta_{\text{ft}}, \theta_b, \{W_i\}, \theta_{\text{out}}\}$. The input feature extractor and readout function can be represented with fully connected neural networks with one or more hidden layers, which we know are universal function approximators, while $\prod_{i=1}^{N} W_i$ corresponds to a set of linear layers. Note that deep ReLU networks act like deep linear networks when the input and pre-synaptic activations are non-negative. We will later show that this is indeed the case within these linear layers, meaning that the neural network function $\hat{f}$ is fully generic and can be represented by deep ReLU networks, as visualized in Figure 1.

Next, we will derive the form of the post-update prediction $\hat{f}(\mathbf{x}^\star; \theta')$. Let $\mathbf{z}_k = \left(\prod_{i=1}^{N} W_i\right)\phi(\mathbf{x}_k)$ and we denote its gradient with respect to the loss as $\nabla_{\mathbf{z}_k} \ell = e(\mathbf{x}_k, \mathbf{y}_k)$. The gradient with respect to any of the weight matrices $W_i$ for a single datapoint $(\mathbf{x}, \mathbf{y})$ is given by

$$\nabla_{W_i} \ell(\mathbf{y}, \hat{f}(\mathbf{x}_k, \theta)) = \left(\prod_{j=1}^{i-1} W_j\right)^T e(\mathbf{x}, \mathbf{y})\phi(\mathbf{x}; \theta_{\text{ft}}, \theta_b)^T \left(\prod_{j=i+1}^{N} W_j\right)^T.$$

Therefore, the post-update value of $\prod_{i=1}^{N} W_i' = \prod_{i=1}^{N}(W_i - \alpha\frac{1}{K}\sum_k \nabla_{W_i})$ is given by

$$\prod_{i=1}^{N} W_i - \frac{\alpha}{K}\sum_{k=1}^{K}\sum_{i=1}^{N}\left(\prod_{j=1}^{i-1} W_j\right)\left(\prod_{j=1}^{i-1} W_j\right)^T e(\mathbf{x}_k, \mathbf{y}_k)\phi(\mathbf{x}_k; \theta_{\text{ft}}, \theta_b)^T\left(\prod_{j=i+1}^{N} W_j\right)^T\left(\prod_{j=i+1}^{N} W_j\right) - O(\alpha^2),$$

where we move the summation over $k$ to the left and where we will disregard the last term, assuming that $\alpha$ is comparatively small such that $\alpha^2$ and all higher order terms vanish. In general, these terms do not necessarily need to vanish, and likely would further improve the expressiveness of the gradient update, but we disregard them here for the sake of the simplicity of the derivation. Ignoring these terms, we now note that the post-update value of $\mathbf{z}^\star$ when $\mathbf{x}^\star$ is provided as input into $\hat{f}(\cdot; \theta')$ is given by

$$\mathbf{z}^\star = \prod_{i=1}^{N} W_i\phi(\mathbf{x}^\star; \theta_{\text{ft}}', \theta_b') \tag{14}$$

$$-\frac{\alpha}{K}\sum_{k=1}^{K}\sum_{i=1}^{N}\left(\prod_{j=1}^{i-1} W_j\right)\left(\prod_{j=1}^{i-1} W_j\right)^T e(\mathbf{x}_k, \mathbf{y}_k)\phi(\mathbf{x}_k; \theta_{\text{ft}}, \theta_b)^T\left(\prod_{j=i+1}^{N} W_j\right)^T\left(\prod_{j=i+1}^{N} W_j\right)\phi(\mathbf{x}^\star; \theta_{\text{ft}}', \theta_b'),$$

and $\hat{f}(\mathbf{x}^\star; \theta') = f_{\text{out}}(\mathbf{z}^\star; \theta'_{\text{out}})$.

Our goal is to show that that there exists a setting of $W_i$, $f_{\text{out}}$, and $\phi$ for which the above function, $\hat{f}(\mathbf{x}^\star, \theta')$, can approximate any function of $(\{(\mathbf{x}, \mathbf{y})_k\}, \mathbf{x}^\star)$. To show universality, we will aim independently control information flow from $\{\mathbf{x}_k\}$, from $\{\mathbf{y}_k\}$, and from $\mathbf{x}^\star$ by multiplexing forward information from $\{\mathbf{x}_k\}$ and $\mathbf{x}^\star$ and backward information from $\{\mathbf{y}_k\}$. We will achieve this by decomposing $W_i$, $\phi$, and the error gradient into three parts, as follows:

$$W_i := \begin{bmatrix} \tilde{W}_i & 0 & 0 \\ 0 & \overline{W}_i & 0 \\ 0 & 0 & \check{w}_i \end{bmatrix} \qquad \phi(\cdot; \theta_{\text{ft}}, \theta_b) := \begin{bmatrix} \tilde{\phi}(\cdot; \theta_{\text{ft}}, \theta_b) \\ \mathbf{0} \\ \theta_b \end{bmatrix} \qquad \nabla_{\mathbf{z}_k} \ell(\mathbf{y}_k, \hat{f}(\mathbf{x}_k; \theta)) := \begin{bmatrix} \mathbf{0} \\ \overline{e}(\mathbf{y}_k) \\ \check{e}(\mathbf{y}_k) \end{bmatrix}$$
(15)

where the initial value of $\theta_b$ will be 0. The top components all have equal numbers of rows, as do the middle components. As a result, we can see that $\mathbf{z}_k$ will likewise be made up of three components, which we will denote as $\tilde{\mathbf{z}}_k$, $\overline{\mathbf{z}}_k$, and $\check{z}_k$. Lastly, we construct the top component of the error gradient to be $\mathbf{0}$, whereas the middle and bottom components, $\overline{e}(\mathbf{y}_k)$ and $\check{e}(\mathbf{y}_k)$, can be set to be any linear (but not affine) function of $\mathbf{y}_k$. We discuss how to achieve this gradient in the latter part of this section when we define $f_{\text{out}}$ and in Section 6.

In Appendix A.3, we show that we can choose a particular form of $\tilde{W}_i$, $\overline{W}_i$, and $\check{w}_i$ that will simplify the products of $W_j$ matrices in Equation 14, such that we get the following form for $\overline{\mathbf{z}}^\star$:

$$\overline{\mathbf{z}}^\star = -\alpha \frac{1}{K} \sum_{k=1}^{K} \sum_{i=1}^{N} A_i \overline{e}(\mathbf{y}_k) \tilde{\phi}(\mathbf{x}_k; \theta_{\text{ft}}, \theta_b)^T B_i^T B_i \tilde{\phi}(\mathbf{x}^\star; \theta_{\text{ft}}, \theta'_b),$$
(16)

where $A_1 = I$, $B_N = I$, $A_i$ can be chosen to be any symmetric positive-definite matrix, and $B_i$ can be chosen to be any positive definite matrix. In Appendix D, we will further show that these definitions of the weight matrices satisfy the condition that their activations are non-negative, meaning that the model $\hat{f}$ can be represented by a generic deep network with ReLU nonlinearities.

Finally, we need to define the function $f_{\text{out}}$ at the output. When a training input $\mathbf{x}_k$ is passed in, we need $f_{\text{out}}$ to propagate information about its corresponding label $\mathbf{y}_k$ as defined in Equation 15. And, when the test input $\mathbf{x}^\star$ is passed in, we need a function defined on $\overline{\mathbf{z}}^\star$. Thus, we will define $f_{\text{out}}$ as a neural network that approximates the following multiplexer function and its derivatives (as shown possible by Hornik et al. (1990)):

$$f_{\text{out}}\left(\begin{bmatrix} \tilde{\mathbf{z}} \\ \overline{\mathbf{z}} \\ \check{z} \end{bmatrix}; \theta_{\text{out}}\right) = \mathbb{1}(\overline{\mathbf{z}} = \mathbf{0}) g_{\text{pre}}\left(\begin{bmatrix} \tilde{\mathbf{z}} \\ \overline{\mathbf{z}} \\ \check{z} \end{bmatrix}; \theta_g\right) + \mathbb{1}(\overline{\mathbf{z}} \neq \mathbf{0}) h_{\text{post}}(\overline{\mathbf{z}}; \theta_h),$$
(17)

where $g_{\text{pre}}$ is a linear function with parameters $\theta_g$ such that $\nabla_{\mathbf{z}} \ell = e(\mathbf{y})$ satisfies Equation 15 (see Section 6) and $h_{\text{post}}(\cdot; \theta_h)$ is a neural network with one or more hidden layers. As shown in Appendix A.4, the post-update value of $f_{\text{out}}$ is

$$f_{\text{out}}\left(\begin{bmatrix} \tilde{\mathbf{z}}^\star \\ \overline{\mathbf{z}}^\star \\ \check{z}^\star \end{bmatrix}; \theta'_{\text{out}}\right) = h_{\text{post}}(\overline{\mathbf{z}}^\star; \theta_h).$$
(18)

Now, combining Equations 16 and 18, we can see that the post-update value is the following:

$$\hat{f}(\mathbf{x}^\star; \theta') = h_{\text{post}}\left(-\alpha \frac{1}{K} \sum_{k=1}^{K} \sum_{i=1}^{N} A_i \overline{e}(\mathbf{y}_k) \tilde{\phi}(\mathbf{x}_k; \theta_{\text{ft}}, \theta_b)^T B_i^T B_i \tilde{\phi}(\mathbf{x}^\star; \theta_{\text{ft}}, \theta'_b); \theta_h\right)$$
(19)

In summary, so far, we have chosen a particular form of weight matrices, feature extractor, and output function to decouple forward and backward information flow and recover the post-update function above. Now, our goal is to show that the above function $\hat{f}(\mathbf{x}^\star; \theta')$ is a universal learning algorithm approximator, as a function of $(\{(\mathbf{x}, \mathbf{y})_k\}, \mathbf{x}^\star)$. For notational clarity, we will use use $k_i(\mathbf{x}_k, \mathbf{x}^\star) := \tilde{\phi}(\mathbf{x}_k; \theta_{\text{ft}}, \theta_b)^T B_i^T B_i \tilde{\phi}(\mathbf{x}^\star; \theta_{\text{ft}}, \theta'_b)$ to denote the inner product in the above equation, noting that it can be viewed as a type of kernel with the RKHS defined by $B_i \tilde{\phi}(\mathbf{x}; \theta_{\text{ft}}, \theta_b)$.[7] The

---

[7] Due to the symmetry of kernels, this requires interpreting $\theta_b$ as part of the input, rather than a kernel hyperparameter, so that the left input is $(\mathbf{x}_k, \theta_b)$ and the right one is $(\mathbf{x}^\star, \theta'_b)$.

connection to kernels is not in fact needed for the proof, but provides for convenient notation and an interesting observation. We can now simplify the form of $\hat{f}(\mathbf{x}^\star, \theta')$ as the following equation:

$$\hat{f}(\mathbf{x}^\star; \theta') = h_{\text{post}}\left(-\alpha\frac{1}{K}\sum_{i=1}^{N}\sum_{k=1}^{K}A_i\overline{e}(\mathbf{y}_k)k_i(\mathbf{x}_k, \mathbf{x}^\star); \theta_h\right) \tag{20}$$

Intuitively, Equation 20 can be viewed as a sum of basis vectors $A_i\overline{e}(\mathbf{y}_k)$ weighted by $k_i(\mathbf{x}_k, \mathbf{x}^\star)$, which is passed into $h_{\text{post}}$ to produce the output. In Appendix C, we show that we can choose $\overline{e}$, $\theta_{\text{ft}}$, $\theta_h$, each $A_i$, and each $B_i$ such that Equation 20 can approximate any continuous function of $(\{(\mathbf{x}, \mathbf{y})_k\}, \mathbf{x}^\star)$ on compact subsets of $\mathbb{R}^{\dim(\mathbf{y})}$. As in the one-shot setting, the bias transformation variable $\theta_b$ plays a vital role in our construction, as it breaks the symmetry within $k_i(\mathbf{x}, \mathbf{x}^\star)$. Without such asymmetry, it would not be possible for our constructed function to represent any function of $\mathbf{x}$ and $\mathbf{x}^\star$ after one gradient step.

In conclusion, we have shown that there exists a neural network structure for which $\hat{f}(\mathbf{x}^\star; \theta')$ is a universal approximator of $f_{\text{target}}(\{(\mathbf{x}, \mathbf{y})_k\}, \mathbf{x}^\star)$.

## C  SUPPLEMENTARY PROOF FOR K-SHOT SETTING

In Section 5 and Appendix B, we showed that the post-update function $\hat{f}(\mathbf{x}^\star; \theta')$ takes the following form:

$$\hat{f}(\mathbf{x}^\star; \theta') = h_{\text{post}}\left(-\alpha\frac{1}{K}\sum_{i=1}^{N}\sum_{k=1}^{K}A_i\overline{e}(\mathbf{y}_k)k_i(\mathbf{x}_k, \mathbf{x}^\star); \theta_h\right)$$

In this section, we aim to show that the above form of $\hat{f}(\mathbf{x}^\star; \theta')$ can approximate any function of $\{(\mathbf{x}, \mathbf{y})_k; k \in 1...K\}$ and $\mathbf{x}^\star$ that is invariant to the ordering of the training datapoints $\{(\mathbf{x}, \mathbf{y})_k; k \in 1...K\}$. The proof will be very similar to the one-shot setting proof in Appendix A.1

Similar to Appendix A.1, we will ignore the first and last elements of the sum by defining $B_1$ to be $\epsilon I$ and $A_N$ to be $\epsilon I$, where $\epsilon$ is a small positive constant to ensure positive definiteness. We will then re-index the first summation over $i = 2...N - 1$ to instead use two indexing variables $j$ and $l$ as follows:

$$\hat{f}(\mathbf{x}^\star; \theta') \approx h_{\text{post}}\left(-\alpha\frac{1}{K}\sum_{j=0}^{J-1}\sum_{l=0}^{L-1}\sum_{k=1}^{K}A_{jl}\overline{e}(\mathbf{y}_k)k_{jl}(\mathbf{x}_k, \mathbf{x}^\star); \theta_h\right)$$

As in Appendix A.1, we will define the function $k_{jl}$ to be an indicator function over the values of $\mathbf{x}_k$ and $\mathbf{x}^\star$. In particular, we will reuse Lemma A.1, which was proved in Appendix A.2 and is copied below:

**Lemma A.1** *We can choose $\theta_{ft}$ and each $B_{jl}$ such that*

$$k_{jl}(\mathbf{x}, \mathbf{x}^\star) := \begin{cases} 1 & \text{if } \text{discr}(\mathbf{x}) = \mathbf{e}_j \text{ and } \text{discr}(\mathbf{x}^\star) = \mathbf{e}_l \\ 0 & \text{otherwise} \end{cases} \tag{8}$$

*where* $\text{discr}(\cdot)$ *denotes a function that produces a one-hot discretization of its input and* $\mathbf{e}$ *denotes the 0-indexed standard basis vector.*

Likewise, we will chose $\overline{e}(\mathbf{y}_k)$ to be the linear function that outputs $J * L$ stacked copies of $\mathbf{y}_k$. Then, we will define $A_{jl}$ to be a matrix that selects the copy of $\mathbf{y}_k$ in the position corresponding to $(j, l)$, i.e. in the position $j + J * l$. This can be achieved using a diagonal $A_{jl}$ matrix with diagonal values of $1 + \epsilon$ at the positions corresponding to the $n$th vector, and $\epsilon$ elsewhere, where $n = (j + J * l)$ and $\epsilon$ is used to ensure that $A_{jl}$ is positive definite.

As a result, the post-update function is as follows:

$$\hat{f}(\mathbf{x}^\star; \theta') \approx h_{\text{post}} \left( -\alpha \frac{1}{K} \sum_{k=1}^{K} v(\mathbf{x}_k, \mathbf{x}^\star, \mathbf{y}_k); \theta_h \right), \text{ where } v(\mathbf{x}, \mathbf{x}^\star, \mathbf{y}) \approx \begin{bmatrix} \mathbf{0} \\ \vdots \\ \mathbf{0} \\ \mathbf{y}_k \\ \mathbf{0} \\ \vdots \\ \mathbf{0} \end{bmatrix},$$

where $\mathbf{y}_k$ is at the position $j + J * l$ within the vector $v(\mathbf{x}_k, \mathbf{x}^\star, \mathbf{y}_k)$, where $j$ satisfies $\text{discr}(\mathbf{x}_k) = \mathbf{e}_j$ and where $l$ satisfies $\text{discr}(\mathbf{x}_k^\star) = \mathbf{e}_l$.

For discrete, one-shot labels $\mathbf{y}_k$, the summation over $v$ amounts to frequency counts of the triplets $(\mathbf{x}_k, \mathbf{x}^\star, \mathbf{y}_k)$. In the setting with continuous labels, we cannot attain frequency counts, as we do not have access to a discretized version of the label. Thus, we must make the assumption that no two datapoints share the same input value $\mathbf{x}_k$. With this assumption, the summation over $v$ will contain the output values $\mathbf{y}_{k'}$ at the index corresponding to the value of $(\mathbf{x}_{k'}, \mathbf{x}^\star)$. For both discrete and continuous labels, this representation is redundant in $\mathbf{x}^\star$, but nonetheless contains sufficient information to decode the test input $\mathbf{x}^\star$ and set of datapoints $\{(\mathbf{x}, \mathbf{y})_k\}$ (but not the order of datapoints).

Since $h_{\text{post}}$ is a universal function approximator and because its input contains all of the information of $(\{(\mathbf{x}, \mathbf{y})_k\}, \mathbf{x}^\star)$, the function $\hat{f}(\mathbf{x}^\star; \theta') \approx h_{\text{post}} \left( -\alpha \frac{1}{K} \sum_{k=1}^{K} v(\mathbf{x}_k, \mathbf{x}^\star, \mathbf{y}_k); \theta_h \right)$ is a universal function approximator with respect to $\{(\mathbf{x}, \mathbf{y})_k\}$ and $\mathbf{x}^\star$.

## D  DEEP RELU NETWORKS

In this appendix, we show that the network architecture with linear layers analyzed in the Sections 4 and 5 can be represented by a deep network with ReLU nonlinearities. We will do so by showing that the input and activations within the linear layers are all non-negative.

First, consider the input $\phi(\cdot; \theta_{\text{ft}}, \theta_b)$ and $\tilde{\phi}(\cdot; \theta'_{\text{ft}}, \theta'_b)$. The input $\tilde{\phi}(\cdot; \theta_{\text{ft}}, \theta_b)$ is defined to consist of three terms. The top term, $\tilde{\tilde{\phi}}$ is defined in Appendices A.2 and C to be a discretization (which is non-negative) both before and after the parameters are updated. The middle term is defined to be a constant $\mathbf{0}$. The bottom term, $\theta_b$, is defined to be $0$ before the gradient update and is not used afterward.

Next, consider the weight matrices, $W_i$. To determine that the activations are non-negative, it is now sufficient to show that the products $W_N, W_{N-1}W_N, ..., \prod_{i=1}^{N} W_i$ are positive semi-definite. To do so, we need to show that the products $\prod_{i=j}^{N} \tilde{W}_i$, $\prod_{i=j}^{N} \overline{W}_i$, and $\prod_{i=j}^{N} \check{w}_i$ are PSD for $j = 1, ..., N$. In Appendix A.2, each $B_i = \tilde{M}_{i+1}$ is defined to be positive definite; and in Appendix A.3, we define the products $\prod_{i=j+1}^{N} \tilde{W}_i = \tilde{M}_j + 1$. Thus, the conditions on the products of $\tilde{W}_i$ are satisfied. In Appendices A.1 and C, each $A_i$ are defined to be symmetric positive definite matrices. In Appendix A.3, we define $\overline{W}_i = \overline{M}_{i-1}^{-1} \overline{M}_i$ where $A_i = \overline{M}_{i-1}\overline{M}_{i-1}^{T}$. Thus, we can see that each $\overline{M}_i$ is also symmetric positive definite, and therefore, each $\overline{W}_i$ is positive definite. Finally, the purpose of the weights $\check{w}_i$ is to provide nonzero gradients to the input $\theta_b$, thus a positive value for each $\check{w}_i$ will suffice.

## E  PROOF OF THEOREM 6.1

Here we provide a proof of Theorem 6.1:

**Theorem 6.1** *The gradient of the standard mean-squared error objective evaluated at $\hat{\mathbf{y}} = \mathbf{0}$ is a linear, invertible function of $\mathbf{y}$.*

For the standard mean-squared error objective, $\ell(\mathbf{y}, \hat{\mathbf{y}}) = \frac{1}{2}\|\mathbf{y} - \hat{\mathbf{y}}\|^2$, the gradient is $\nabla_{\hat{\mathbf{y}}}\ell(\mathbf{y}, \mathbf{0}) = -\mathbf{y}$, which satisfies the requirement, as $A = -I$ is invertible.

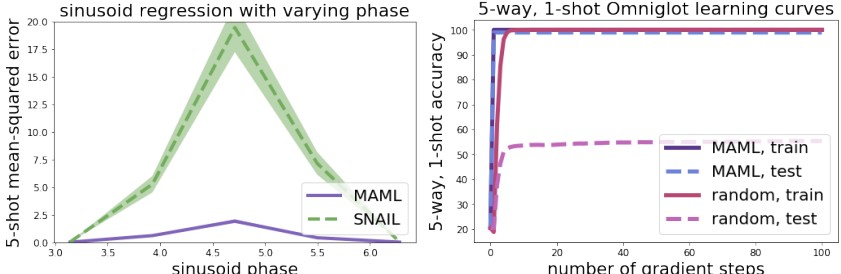

Figure 6: Left: Another comparison with out-of-distribution tasks, varying the phase of the sine curve. There is a clear trend that gradient descent enables better generalization on out-of-distribution tasks compared to the learning strategies acquired using recurrent meta-learners such as SNAIL. Right: Here is another example that shows the resilience of a MAML-learned initialization to overfitting. In this case, the MAML model was trained using one inner step of gradient descent on 5-way, 1-shot Omniglot classification. Both a MAML and random initialized network achieve perfect training accuracy. As expected, the model trained from scratch catastrophically overfits to the 5 training examples. However, the MAML-initialized model does not begin to overfit, even after 100 gradient steps.

## F    PROOF OF THEOREM 6.2

Here we provide a proof of Theorem 6.2:

**Theorem 6.2** *The gradient of the softmax cross entropy loss with respect to the pre-softmax logits is a linear, invertible function of* **y***, when evaluated at* **0***.*

For the standard softmax cross-entropy loss function with discrete, one-hot labels $\mathbf{y}$, the gradient is $\nabla_{\hat{\mathbf{y}}}\ell(\mathbf{y}, \mathbf{0}) = \mathbf{c} - \mathbf{y}$ where $\mathbf{c}$ is a constant vector of value $c$ and where we are denoting $\hat{\mathbf{y}}$ as the pre-softmax logits. Since $\mathbf{y}$ is a one-hot representation, we can rewrite the gradient as $\nabla_{\hat{\mathbf{y}}}\ell(\mathbf{y}, \mathbf{0}) = (C - I)\mathbf{y}$, where $C$ is a constant matrix with value $c$. Since $A = C - I$ is invertible, the cross entropy loss also satisfies the above requirement. Thus, we have shown that both of the standard supervised objectives of mean-squared error and cross-entropy allow for the universality of gradient-based meta-learning.

## G    ADDITIONAL EXPERIMENTAL DETAILS

In this section, we provide two additional comparisons on an out-of-distribution task and using additional gradient steps, shown in Figure 6. We also include additional experimental details.

### G.1    INDUCTIVE BIAS EXPERIMENTS

For Omniglot, all meta-learning methods were trained using code provided by the authors of the respective papers, using the default model architectures and hyperparameters. The model embedding architecture was the same across all methods, using 4 convolutional layers with $3 \times 3$ kernels, 64 filters, stride 2, batch normalization, and ReLU nonlinearities. The convolutional layers were followed by a single linear layer. All methods used the Adam optimizer with default hyperparameters. Other hyperparameter choices were specific to the algorithm and can be found in the respective papers. For MAML in the sinusoid domain, we used a fully-connected network with two hidden layers of size 100, ReLU nonlinearities, and a bias transformation variable of size 10 concatenated to the input. This model was trained for 70,000 meta-iterations with 5 inner gradient steps of size $\alpha = 0.001$. For SNAIL in the sinusoid domain, the model consisted of 2 blocks of the following: 4 dilated convolutions with $2 \times 1$ kernels 16 channels, and dilation size of 1,2,4, and 8 respectively, then an attention block with key/value dimensionality of 8. The final layer is a $1 \times 1$ convolution to the output. Like MAML, this model was trained to convergence for 70,000 iterations using Adam with default hyperparameters. We evaluated the MAML and SNAIL models for 1200 trials, reporting the mean and $95\%$ confidence intervals. For computational reasons, we evaluated the MetaNet model using 600 trials, also reporting the mean and $95\%$ confidence intervals.

Following prior work (Santoro et al., 2016), we downsampled the Omniglot images to be $28 \times 28$. When scaling or shearing the digits to produce out-of-domain data, we transformed the original $105 \times 105$ Omniglot images, and then downsampled to $28 \times 28$.

## G.2 DEPTH EXPERIMENTS

In the depth comparison, all models were trained to convergence using 70,000 iterations. Each model was defined to have a fixed number of hidden units based on the total number of parameters (fixed at around 40,000) and the number of hidden layers. Thus, the models with 2, 3, 4, and 5 hidden layers had 200, 141, 115, and 100 units per layer respectively. For the model with 1 hidden layer, we found that using more than $20,000$ hidden units, corresponding to $40,000$ parameters, resulted in poor performance. Thus, the results reported in the paper used a model with 1 hidden layer with $250$ units which performed much better. We trained each model three times and report the mean and standard deviation of the three runs. The performance of an individual run was computed using the average over

