# OpenReview forum: "Meta-Learning and Universality: Deep Representations and Gradient Descent can Approximate any Learning Algorithm"
_ICLR.cc/2018/Conference — Accept (Poster)_

### Official Review · AnonReviewer3 · 2017-11-27
**Technically interesting work but practical significance seems highly questionable**

**Rating:** 6
**Confidence:** 3

**Review:**

This paper studies the capacity of the model-agnostic meta-learning (MAML) framework as a universal learning algorithm approximator. Since a (supervised) learning algorithm can be interpreted as a map from a dataset and an input to an output, the authors define a universal learning algorithm approximator to be a universal function approximator over the set of functions that map a set of data points and an  input to an output. The authors show constructively that there exists a neural network architecture for which the model learned through MAML can approximate any learning algorithm.

The paper is for the most part clear, and the main result seems original and technically interesting. At the same time, it is not clear to me that this result is also practically significant. This is because the universal approximation result relies on a particular architecture that is not necessarily the design one would always use in MAML. This implies that MAML as typically used (including in the original paper by Finn et al, 2017a) is not necessarily a universal learning algorithm approximator, and this paper does not actually justify its empirical efficacy theoretically. For instance, the authors do not even use the architecture proposed in their proof in their experiments. This is in contrast to the classical universal function approximator results for feedforward neural networks, as a single hidden layer feedforward network is often among the family of architectures considered in the course of hyperparameter tuning. This distinction should be explicitly discussed in the paper.  Moreover, the questions posed in the experimental results do not seem related to the theoretical result, which seems odd.

Specific comments and questions:
Page 4: "\hat{f}(\cdot; \theta') approximates f_{\text{target}}(x, y, x^*) up to arbitrary position". There seems to be an abuse of notation here as the first expression is a function and the second expression is a value.
Page 4: "to show universality, we will construct a setting of the weight matrices that enables independent control of the information flow...". How does this differ from the classical UFA proofs? The relative technical merit of this paper would be more clear if this is properly discussed.
Page 4: "\prod_{i=1}^N (W_i - \alpha \nabla_{W_i})". There seems to be a typo here: \nabla_{W_i} should be \nabla_{W_i} L.
Page 7: "These error functions effectively lose information because simply looking at their gradient is insufficient to determine the label." It would be interesting the compare the efficacy of MAML on these error functions as compared to cross entropy and mean-squared error.
Page 7: "(1) can a learner trained with MAML further improve from additional gradient steps when learning new tasks at test time...? (2) does the inductive bias of gradient descent enable better few-shot learning performance on tasks outside of the training distribution...?". These questions seem unrelated to the universal learning algorithm approximator result that constitutes the main part of the paper. If you're going to study these question empirically, why didn't you also try to investigate them theoretically (e.g. sample complexity and convergence of MAML)? A systematic and comprehensive analysis of these questions from both a theoretical and empirical perspective would have constituted a compelling paper on its own.
Pages 7-8: Experiments. What are the architectures and hyperparameters used in the experiments, and how sensitive are the meta-learning algorithms to their choice?
Page 8: "our experiments show that learning strategies acquired with MAML are more successful when faced with out-of-domain tasks compared to recurrent learners....we show that the representations acquired with MAML are highly resilient to overfitting". I'm not sure that such general claims are justified based on the experimental results in this paper. Generalizing to out-of-domain tasks is heavily dependent on the specific level and type of drift between the old and new distributions. These properties aren't studied at all in this work.


POST AUTHOR REBUTTAL: After reading the response from the authors and seeing the updated draft, I have decided to upgrade my rating of the manuscript to a 6. The universal learning algorithm approximator result is a nice result, although I do not agree with the other reviewer that it is a  "significant contribution to the theoretical understanding of meta-learning," which the authors have reinforced (although it can probably be considered a significant contribution to the theoretical understanding of MAML in particular). Expressivity of the model or algorithm is far from the main or most significant consideration in a machine learning problem, even in the standard supervised learning scenario. Questions pertaining to issues such as optimization and model selection are just as, if not more, important. These sorts of ideas are explored in the empirical part of the paper, but I did not find the actual experiments in this section to be very compelling. Still, I think the universal learning algorithm approximator result is sufficient on its own for the paper to be accepted.

---

> ### Author Response · Authors · 2017-12-19
> **Revised paper, addressing concerns. [part 1/2]**
>
> Thank you for the constructive feedback. All of the concerns raised in the review have been addressed in the revised version of the paper.
>
> Please see our main response in a comment above that addresses the primary concerns among all reviewers. We reply to your specific comments here.
>
> > “...This is because the universal approximation result relies on a particular architecture that is not necessarily the design one would always use in MAML. ... For instance, the authors do not even use the architecture proposed in their proof in their experiments...”
> As mentioned above, we would like to clarify that the result holds for a generic deep network with ReLU nonlinearities that is used in prior papers that use MAML [Finn et al. ‘17ab, Reed et al. ‘17] and in the experiments in Section 7 of this paper. We revised Section 4 and Appendix D of the paper to make this more clear and explicitly show how this is the case.
>
> > “Page 4: "\hat{f}(\cdot; \theta') approximates f_{\text{target}}(x, y, x^*) up to arbitrary position". There seems to be an abuse of notation here as the first expression is a function and the second expression is a value.”
> > “Page 4: "\prod_{i=1}^N (W_i - \alpha \nabla_{W_i})". There seems to be a typo here: \nabla_{W_i} should be \nabla_{W_i} L.”
> Thank you for catching these two typos. We fixed both.
>
> > Page 4: "to show universality, we will construct a setting of the weight matrices that enables independent control of the information flow...". How does this differ from the classical UFA proofs? The relative technical merit of this paper would be more clear if this is properly discussed.
> We added text in the latter part of section 3 to clarify the relationship to the UFA theorem: “It is clear how $f_\text{MAML}$ can approximate any function on $x^\star$, as per the UFA theorem; however, it is not obvious if $f_\text{MAML}$ can represent any function of the set of input, output pairs in $\dataset_\task$, since the UFA theorem does not consider the gradient operator.”
> Our proof uses the UFA proof as a subroutine, and is otherwise completely distinct.
>
> > “These questions seem unrelated to the universal learning algorithm approximator result that constitutes the main part of the paper. If you're going to study these question empirically, why didn't you also try to investigate them theoretically (e.g. sample complexity and convergence of MAML)? A systematic and comprehensive analysis of these questions from both a theoretical and empirical perspective would have constituted a compelling paper on its own.”
> Yes, these two questions would be very interesting to analyze theoretically. We leave such theoretical questions to future work. With regard to the connection between these experiments and the theory, please see our comment above to all of the reviewers -- we added another experiment in Section 7.2 which directly follows up on the theory, studying the depth necessary to meta-learn a distribution of tasks compared to the depth needed for standard learning. We also added more discussion connecting the theory and the existing experiments.
>
> > “What are the architectures and hyperparameters used in the experiments, and how sensitive are the meta-learning algorithms to their choice?”
> We outlined most of the experimental details in the main text and in the Appendix. We added some additional details that we had missed, in Sections 7.1 and Appendix G.
> Omniglot:
> We use a standardized convolutional encoder architecture in the Omniglot domain (4 conv layers each with 64 3x3 filters, stride 2, ReLUs, and batch norm, followed by a linear layer). All methods used the Adam optimizer with default hyperparameters. Other hyperparameter choices were specific to the algorithm and can be found in the respective papers.
> Sinusoid:
> With MAML, we used a simple fully-connected network with 2 hidden layers of width 100 and ReLU nonlinearities, and the suggested hyperparameters in the MAML codebase (Adam optimizer, alpha=0.001, 5 gradient steps). On the sinusoid task with TCML, we used an architecture of 2x{ 4 dilated convolution layers with 16 channels, 2x1 kernels, and dilation size of 1,2,4,8 respectively; then an attention block with key/value dimensionality of 8} followed by a 1x1 conv. TCML used the Adam optimizer with default hyperparameters.
> We have not found any of the algorithms to be particularly sensitive to the architecture or hyperparameters. The hyperparameters provided in each paper’s codebases worked well.

---

> > ### Author Response · Authors · 2017-12-20
> > **Revised paper, addressing concerns. [part 2/2]**
> >
> > > “I'm not sure that such general claims are justified based on the experimental results in this paper. Generalizing to out-of-domain tasks is heavily dependent on the specific level and type of drift between the old and new distributions. These properties aren't studied at all in this work.”
> > We modified the first-mentioned claim to be more precise. We agree that out-of-domain generalization is heavily dependent on both the task and the form of drift. Thus, we aimed to study many different levels and types of drift, studying four different types of drift (shear, scale, amplitude, phase) and several levels/amounts of each of these types of drift, within two different problem domains (Omniglot, sinusoid regression). In every single type and level of drift that we experimented with, we observed the same result -- that gradient-descent generalized better than recurrent networks.
> > With regard to the second claim on resilience to overfitting, this claim is in the context of the experiments with additional gradient steps and is not referring to out-of-domain tasks. The claim is supported by the results in our experiments.

---

### Official Review · AnonReviewer2 · 2017-11-27
**Result looks interesting. Presentation could be further improved.**

**Rating:** 6
**Confidence:** 1

**Review:**

The paper tries to address an interesting question: does deep representation combined with standard gradient descent have sufficient capacity to approximate any learning algorithm. The authors provide answers, both theoretically and empirically.

The presentation could be further improved. For example,

-the notation $\mathcal{L}$ is inconsistent. It has different inputs at each location.
-the bottom of page 5, "we then define"?
-I couldn't understand the sentence "can approximate any continuous function of (x,y,x^*) on compact subsets of R^{dim(y)}" in Lemma 4.1".
-before Equation (1), "where we will disregard the last term.." should be further clarified.
-the paragraph before Section 4. "The first goal of this paper is to show that f_{MAML} is a universal function approximation of (D_{\mathcal{T}},x^*)"? A function can only approximate the same type function.

---

> ### Author Response · Authors · 2017-12-19
> **Revised paper addressing comments**
>
> Please see our main response in a comment above that addresses the primary concerns among all reviewers. We reply to your specific comments here.
>
> >"the notation $\mathcal{L}$ is inconsistent. It has different inputs at each location"
> Thank you for pointing this out. We have modified the paper in Sections 2.2, 3, and 4 to use two different symbols and use each of these symbols in a consistent manner.
>
> >"-the bottom of page 5, "we then define"?"
> The lemma previously appeared on the following page, after “we then define”. Now, it appears on the same page.
>
> > "I couldn't understand the sentence "can approximate any continuous function of (x,y,x^*) on compact subsets of R^{dim(y)}" in Lemma 4.1". "
> We added a footnote to clarify that this assumption is inherited from the UFA theorem.
>
> > the paragraph before Section 4. "The first goal of this paper is to show that f_{MAML} is a universal function approximation of (D_{\mathcal{T}},x^*)"? A function can only approximate the same type function.
> We modified to text at the end of Section 3 to make it clear that f_{MAML} is the same type of function.

---

### Official Review · AnonReviewer1 · 2017-11-27
**Review (educated guess)**

**Rating:** 7
**Confidence:** 1

**Review:**

The paper provides proof that gradient-based meta-learners (e.g. MAML) are "universal leaning algorithm approximators".

Pro:
- Generally well-written with a clear (theoretical) goal
- If the K-shot proof is correct*, the paper constitutes a significant contribution to the theoretical understanding of meta-learning.
- Timely and relevant to a large portion of the ICLR community (assuming the proofs are correct)

Con:
- The theoretical and empirical parts seem quite disconnected. The theoretical results are not applied nor demonstrated in the empirical section and only functions as an underlying premise. I wonder if a purely theoretical contribution would be preferable (or with even fewer empirical results).

* It has not yet been possible for me to check all the technical details and proofs.

---

> ### Author Response · Authors · 2017-12-19
> **New experiment & more discussion added**
>
> Please see our main response in a comment above that addresses the primary concerns among all reviewers. We reply to your specific comments here.
>
> > “The theoretical and empirical parts seem quite disconnected.”
> As mentioned in our main response above, we added a new experiment in Section 7.2 that connects to the theory. The theory suggests that depth is important for an expressive meta-learner compared to standard neural network learner, for which a single hidden layer should theoretically suffice. The results in our new experimental analysis support our theoretical finding that more depth is needed for MAML than for representing individual tasks. We also added additional discussion to clarify and motivate the existing experiments of inductive bias.

---

### Author Response · Authors · 2017-12-19
**Main response to reviewers**

We thank the reviewers for their constructive feedback!

We would first like to clarify that the main theoretical result holds for a generic deep network with ReLU nonlinearities, an architecture which is standard in practice. We have revised Section 4 and Appendix D in the paper to clarify and explicitly show this. As mentioned by R1, this theoretical result is a “significant contribution to the theoretical understanding of meta-learning”.

Second, to address the reviewers concerns about a disconnect between the theory and experiments, we did two things:
1) We added a new experiment in Section 7.2 that directly follows up on the theoretical result, empirically comparing the depth required for meta-learning to the depth required for representing the individual tasks being meta-learned. The empirical results in this section support the theoretical result.
2) We clarified in Section 7 the importance of the existing experiments, which is as follows: the theory shows that MAML is just as expressive as black-box (e.g. RNN-based) meta-learners, but this does not, by itself, indicate why we might prefer one method over the other and in which cases we should prefer one over the other. The experiments illustrate how MAML can improve over black-box meta-learners when extrapolating to out-of-distribution tasks.

We respond to individual comments in direct replies to the reviewers comments. Given the low confidence scores, we hope that the reviewers will follow up on our response and adjust their reviews based on our response if things have become more clear.

---

### Public Comment · (anonymous) · 2018-01-04
**Important paper**

I want to thank the authors for preparing the paper.
The paper clearly shows that model-agnostic meta-learning (MAML) can approximate any learning algorithm.
This was not obvious to me before.

I have now more confidence to apply MAML on many new tasks.

---

### Decision · Program_Chairs · 2018-01-29
**ICLR 2018 Conference Acceptance Decision**

**Decision:**

Accept (Poster)

**Comment:**

R3 summarizes the reasons for the decision on this paper: "The universal learning algorithm approximator result is a nice result, although I do not agree with the other reviewer that it is a  "significant contribution to the theoretical understanding of meta-learning," which the authors have reinforced (although it can probably be considered a significant contribution to the theoretical understanding of MAML in particular). Expressivity of the model or algorithm is far from the main or most significant consideration in a machine learning problem, even in the standard supervised learning scenario. Questions pertaining to issues such as optimization and model selection are just as, if not more, important. These sorts of ideas are explored in the empirical part of the paper, but I did not find the actual experiments in this section to be very compelling. Still, I think the universal learning algorithm approximator result is sufficient on its own for the paper to be accepted."